# Nanowire-based smart windows combining electro- and thermochromics for dynamic regulation of solar radiation

Si-Zhe Sheng[1,5], Jin-Long Wang[2,5], Bin Zhao[3,5], Zhen He[2], Xue-Fei Feng[1], Qi-Guo Shang[1], Cheng Chen[1], Gang Pei[3], Jun Zhou[4], Jian-Wei Liu ●[1] ✉ & Shu-Hong Yu ●[1,2] ✉

Smart window is an attractive option for efficient heat management to minimize energy consumption and improve indoor living comfort owing to their optical properties of adjusting sunlight. To effectively improve the sunlight modulation and heat management capability of smart windows, here, we propose a co-assembly strategy to fabricate the electrochromic and thermochromic smart windows with tunable components and ordered structures for the dynamic regulation of solar radiation. Firstly, to enhance both illumination and cooling efficiency in electrochromic windows, the aspect ratio and mixed type of Au nanorods are tuned to selectively absorb the near-infrared wavelength range of 760 to 1360 nm. Furthermore, when assembled with electrochromic $W_{18}O_{49}$ nanowires in the colored state, the Au nanorods exhibit a synergistic effect, resulting in a 90% reduction of near-infrared light and a corresponding 5 °C cooling effect under 1-sun irradiation. Secondly, to extend the fixed response temperature value to a wider range of 30–50 °C in thermochromic windows, the doping amount and mixed type of W-$VO_2$ nanowires are carefully regulated. Last but not the least, the ordered assembly structure of the nanowires can greatly reduce the level of haze and enhance visibility in the windows.

Nowadays, the energy consumption required to improve occupants' comfort accounts for approximately 15–20% of global energy usage, a figure which is expected to increase exponentially in the coming decades[1,2]. In particular, the overreliance on air conditioning to achieve comfortable indoor temperatures results in significant power consumption and corresponding greenhouse gas emissions[3]. Research in this field has shown that installing windows with reversible optical properties can result in a 50% reduction in energy consumption for cooling, heating, and lighting in buildings[4–7]. The effectiveness of daylighting in buildings relies heavily on the visible light spectrum (VIS, 400–760 nm). The remaining 50% of solar radiation is primarily located in the near-infrared region (NIR, 760–2500 nm), and is mostly utilized for interior heating purposes[8,9]. The ability to modulate the NIR transmission through the windows has a significant effect on regulating indoor living temperature and reducing energy saving. It's worth noting that a mere 1 °C change in

[1]Department of Chemistry, New Cornerstone Science Laboratory, Institute of Biomimetic Materials & Chemistry, Anhui Engineering Laboratory of Biomimetic Materials, Division of Nanomaterials & Chemistry, Hefei National Research Center for Physical Sciences at the Microscale, University of Science and Technology of China, 230026 Hefei, Anhui, China. [2]Institute of Innovative Materials (I2M), Department of Materials Science and Engineering, Southern University of Science and Technology, 518055 Shenzhen, China. [3]Department of Thermal Science and Energy Engineering, University of Science and Technology of China, 230026 Hefei, Anhui, China. [4]Hefei National Research Center for Physical Sciences at the Microscale, University of Science and Technology of China, 230026 Hefei, Anhui, China. [5]These authors contributed equally: Si-Zhe Sheng, Jin-Long Wang, Bin Zhao. ✉e-mail: jwliu13@ustc.edu.cn; shyu@ustc.edu.cn

indoor temperature can result in a 10% reduction in energy consumption[10,11].

As a viable technology to economize building energy consumption, smart windows based on electrochromic[12,13] and thermochromic materials[14,15] have been widely reported due to the reversible optical switching. Recently, these smart windows exhibit excellent modulation efficiency by switching rapidly from bleached to colored state, allowing for uniform tinting across the entire window surface in mere seconds[16,17]. Furthermore, various micro- and nano-structures are used for improving the performance of these materials to meet different applications[18–20]. These previous works demonstrated the potential of smart windows to realize solar radiation modulation and energy saving. However, most of them only achieve the regulation of solar radiation through the transparent and opaque states of smart windows, without intelligent control over specific bands of sunlight that buildings require based on individual needs[21–23]. Higher visible light transmission is critical for indoor lighting but is often sacrificed for higher modulation efficiency[24], and the limited stimulus responsiveness makes it hard to modulate solar radiation to deal with complex weather changes and personal preferences[25]. It is therefore intellectually and practically rewarding to develop smart windows that can effectively block solar radiation in response to environmental changes.

In this work, we propose a general and feasible co-assembly strategy to fabricate smart windows for solar regulation by adjusting the muti-component and assembly structure. By integrating tiny amounts of plasmonic Au nanorods (NRs) with electrochromic $W_{18}O_{49}$ nanowires (NWs), the optical modulation performance of conventional electrochromic windows has been significantly improved. A mixture of Au NRs with varying aspect ratios exhibits strong absorption within the specific wavelength range of 760–1360 nm, allowing for the blocking of over 50% of light within this band. When a negative voltage of 1.5 V is applied, the device translates to a colored state that can effectively block the majority of near-infrared light (over 90%), resulting in a temperature reduction of approximately 5 °C indoors under 1 sun irradiation (maximum natural sunlight). Significantly, the ordered arrangement of these nanomaterials reduces the haze of windows while still maintaining visible light transmittance of 70%. This strategy of multi-component modulation in devices can also be applied to improve the performance of traditional thermochromic windows.

Compared with the fixed phase change temperature of $VO_2$ ($T_c = 68$ °C), the thermochromic W-$VO_2$ NWs with different amounts of tungsten (W) doping were co-assembled to broaden the temperature stimulus response of smart windows. These smart windows, which have a wide response range of 30–50 °C, can dynamically adjust their blocking performance. As the ambient temperature increases, the window's ability to block sunlight becomes stronger. Moreover, the simplicity of the co-assembly method enables the easy fabrication of large-area smart windows with dimensions of $25 \times 20$ cm$^2$, highlighting the potential for scalable production and promising real-world applications in diverse weather conditions.

## Results

### Preparation and regulation of nanowire-based smart windows

The interface co-assembly method is a versatile and effective approach to preparing smart windows with tunable properties, which involves the simultaneous assembly of multiple nanomaterials at an interface to create a functional composite structure[26,27]. Figure 1a illustrates the fabrication process and regulation mechanism of nanowire-based smart windows. To create a selective light absorption electrochromic (SLE) smart window, different aspect ratios of Au NRs with narrow absorption ranges are synthesized and mixed to cover the 760–1360 nm band. Then, the multi-sized Au NR mixture, along with electrochromic $W_{18}O_{49}$ NWs and conductive Ag NWs, are co-assembled into an ordered network structure for precise modulation of the NIR region. This window is capable of selectively absorbing near-infrared wavelength range of 760–1360 nm for both illumination and cooling efficiency, and the synergy of these optical materials can significantly improve the cooling effect at higher temperatures. Using the same interfacial assembly regulation strategy, a wide response-range thermochromic (WRT) smart window was developed by co-assembling thermochromic W-$VO_2$ NWs with different amounts of W doping, extending the fixed response temperature value to the wide temperature response range. Compared to the narrow temperature response range of thermochromic windows using a single type of $VO_2$ NWs, the WRT smart window uses W-$VO_2$ NWs with different phase transition temperatures to achieve a wide temperature response range of 30–50 °C. This is because only the nanowires with high doping amounts undergo a phase change at low temperature, while several

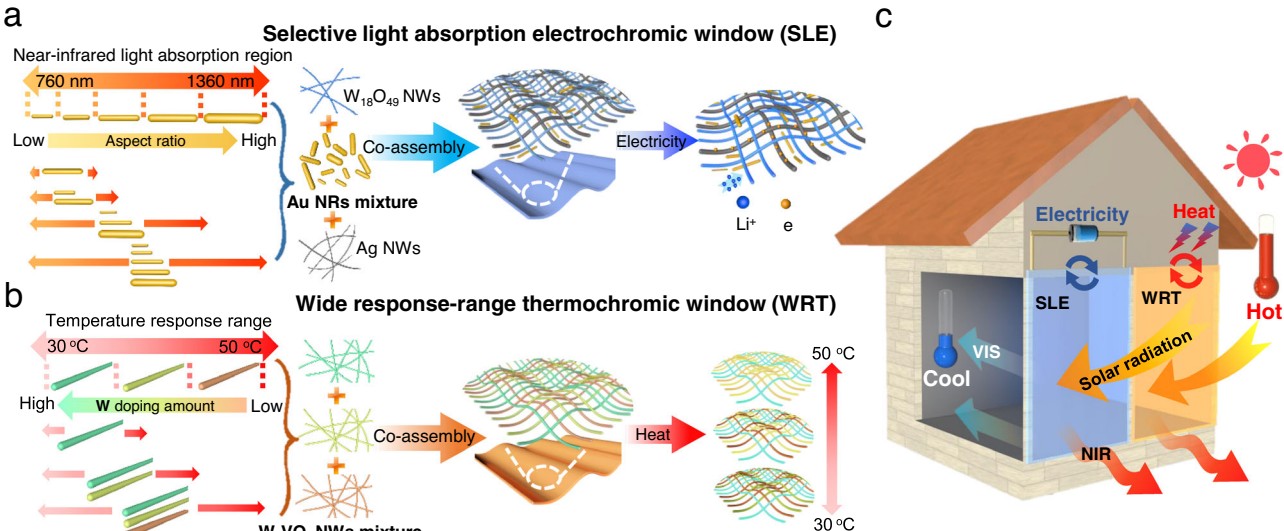

**Fig. 1 | The schematic illustration for the fabrication and modulation mechanism of smart windows. a** The preparation strategy of selective light absorption electrochromic smart window based on co-assembly of multiple nanowires and Au nanorods. **b** The preparation strategy of wide-range thermochromic smart window based on co-assembly of $VO_2$ nanowires with different W doping amounts. **c** The working effect of the house equipped with these smart windows when applied with a small voltage or ambient temperature change.

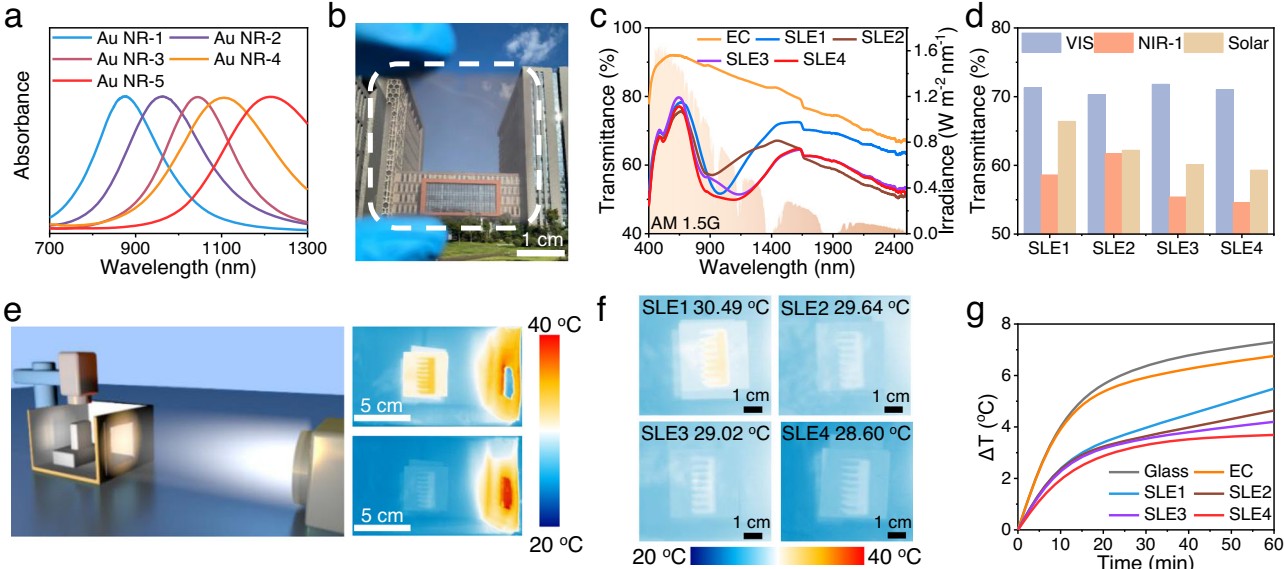

**Fig. 2 | Selective light absorption of smart windows. a** Normalized absorbance spectra of Au NRs with multiple sizes. **b** Photographs of the selective light absorption-electrochromic film. **c** The transmittance spectra of electrochromic film and SLE films under solar spectral irradiance (AM 1.5G). **d** The transmitted irradiance of SLE films in the VIS, NIR−1, and solar regions. **e** The indoor cooling performance measure apparatus for films under the 100 mW cm⁻² illumination to simulate sunlight. An IR camera was used to real-time record the temperature of the blackbody in the center of model chamber to demonstrate the cooling performance of smart windows. **f, g** The infrared images and temperature change for blackbody in the model chamber installed with different films under the 100 mW cm⁻² simulated illumination for 1 h. Source data are provided as a Source Data file.

nanowires with different doping amounts are in the phase transition state as the temperature increases. This broad response range enables smart windows to gradually modulate their light-blocking performance as the ambient temperature varies, providing effective control over indoor lighting and temperature (Fig. 1b). As shown in Fig. 1c, when the smart windows are installed in a house, they selectively block solar radiation and dynamically regulate the indoor temperature in response to a small applied voltage or ambient temperature change.

## Passive regulation of SLE smart windows

The Au NRs with localized surface plasmon resonance (LSPR) effect can only absorb light of specific wavelengths due to determined morphology[28,29]. To achieve selective absorption of NIR light, especially the band that occupies three-quarters of the energy in the NIR-1 region (760–1360 nm). Au NRs with different aspect ratios (Supplementary Figs. 1, 2) were synthesized by regulating the amount of seed solution and the pH of the growth solution (Supplementary Table 1). When the aspect ratio of Au NRs increases from 4 to 9, their absorption peak shows a redshift from 842 to 1212 nm (Fig. 2a). Multiple sizes of Au NRs were mixed and co-assembled with Ag and $W_{18}O_{49}$ NWs (Supplementary Fig. 3). And the $W_{18}O_{49}$/Ag and $W_{18}O_{49}$/Au monolayers were deposited multiple times by Langmuir-Blodgett (LB) co-assembly technique to obtain the network nanowire structure with vertical intersection angles (Supplementary Figs. 4, 5). The conductive layer was created using Ag NWs instead of traditional rigid ITO conductive glass, while ultrafine electrochromic $W_{18}O_{49}$ NWs were utilized to separate Au NRs and Ag NWs, thereby ensuring visible light transmittance, and to further block solar radiation in the colored state.

To verify the superiority of the orderly cross-aligned structures, we also randomly sprayed the same amount of materials onto the substrate to form a disordered structure. The resultant orderly cross-aligned structure can significantly decrease the scattering of light in all directions, resulting in a decrease in film haze from 37.7% of disordered structure to 14% (Supplementary Fig. 6). According to Supplementary Table 2, different layers of $W_{18}O_{49}$/Ag NWs can be transferred (Supplementary Fig. 7a) onto either a flexible polycarbonate (PC) or a rigid

glass substrate. The average transmittance of the films can be tuned from 96.2% to 77.6% at visible light and from 95.1% to 79.3% at NIR-1 (Supplementary Fig. 7b), The corresponding sheet resistance decreases from 826 to 15 Ω sq⁻¹ (Supplementary Fig. 7c). Taking into account the requirements for optical property and conductivity, the 6 layers of $W_{18}O_{49}$ NW networks and 4 layers of $W_{18}O_{49}$/Ag NW networks were laminated to form the electrochromic (EC) film (Supplementary Fig. 8). In contrast, 6 layers of $W_{18}O_{49}$/Au networks and 4 layers of $W_{18}O_{49}$/Ag NW networks were stacked to form the SLE film (Fig. 2b).

Different types and amounts of Au NRs were mixed in SLE films to match the solar radiation spectrum (Supplementary Table 3). All SLE films display well-visible transmittance as high as 70% and good imaging fidelity (Supplementary Fig. 9), while the transmittance of the films could be tuned from 61.7% to 54.6% in NIR-1 (Fig. 2c and Supplementary Table 4). With the types of Au NRs increasing, the absorption range of the near-infrared range becomes wider to cover the entire NIR-1 region (Fig. 2d). To further optimize the performance of films, different layers of $W_{18}O_{49}$/Au were transferred onto the 4 layers of $W_{18}O_{49}$/Ag NW networks in a mutually perpendicular direction. As the number of layers of $W_{18}O_{49}$/Au increases from 2 to 12, the content of Au NRs increases, and the film displays a light red color. The film with 8 layers of $W_{18}O_{49}$/Au still maintains 70% in visible light and drops to 48.1% in NIR-1 (Supplementary Fig. 10). As the number of layers of $W_{18}O_{49}$/Au increases continuously to 10 and 12 layers, their transmittance is further reduced to 43.8% and 41.2% in NIR-1, while remains 65.6% and 59.8% at the visible light region, respectively (Supplementary Table 5).

Moreover, to investigate the temperature regulation performance of the fabricated EC and SLE films, A $3.0 \times 3.0 \times 1.5$ cm³ black anodized cube was utilized as the blackbody absorber in the center of the model chamber. As shown in Fig. 2e, the model chamber with a 10 cm² window was exposed to 1 sun illumination (100 mW cm⁻²). An IR camera was used to real-time record the temperature of the blackbody to demonstrate the performance of windows. Under direct illumination for 1 h, the temperature of the blackbody in the chamber with SLE1-4 films reaches 30.49 °C, 29.64 °C, 29.02 °C, and 28.6 °C, respectively (Supplementary Fig. 11). The temperature of the blackbody in the

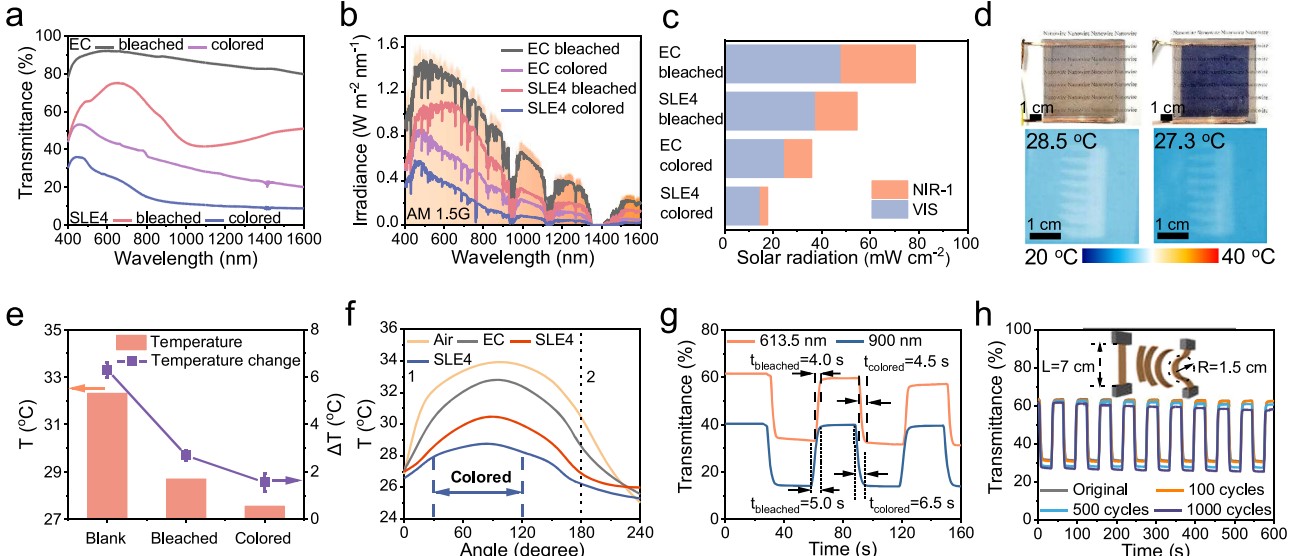

**Fig. 3 | Dynamic regulation of solar radiation through electrochromic function.** **a** The transmittance spectra of EC and SLE4 at bleached and colored states. **b**, **c** The transmitted solar irradiance of EC and SLE4 films at bleached and colored states under solar spectral irradiance. **d** Photographs of SLE4 smart window solid devices at bleached and colored state with the infrared images for the indoor blackbody under the 100 mW cm$^{-2}$ illumination. **e** The temperature of blackbody in the model chamber installed with bleached and colored selective SLE4 smart window solid devices under the simulated sunlight. The corresponding error bar represents the standard deviation. For calculation, the cooling effect of each sample was measured

three times. **f** Measured temperature variation of indoor blackbody and outdoor air in the SLE4 smart window chamber and EC window chamber under simulated illumination of different directions. The windows were exposed to simulated light in area 1 and out of light in area 2. **g** The electrochromic switching time of SLE4 film. The bleached time ($t_{bleached}$) is 4.0 s and colored time ($t_{colored}$) is 4.5 s at 613.5 nm, and the bleached time is 5.0 s and colored time is 6.5 s at 900 nm. **h** In situ transmittance spectra of SLE4 film after bending cycle tests. The length of the sample is 7 cm, and the bending radius of the sample is 1.5 cm. Source data are provided as a Source Data file.

model chamber with normal glass and EC film reached 32.3 °C and 31.8 °C (Fig. 2f), indicating the SLE4 films reduce the temperature as high as 3.7 °C (Fig. 2g).

## Active regulation of SLE smart windows

Except for the continuous light absorption of Au NRs, the electrochromic W$_{18}$O$_{49}$ NWs can further block solar radiation in colored state. EC and SLE4 films can switch between colored and bleached states when applied with a voltage (versus Ag/AgCl) from −1.0 V to +0.2 V in 1 M LiClO$_4$/polycarbonate electrolyte solution (Supplementary Fig. 12). Figure 3a, b show the transmittance spectra of EC and SLE4 films at the bleached and colored states, the colored EC window has 46.3% transmittance at visible light and 30.5% in NIR-1 (Supplementary Table 6). As for colored SLE4 film, nearly 30% visible light, and only 9.1% of NIR-1 are transmitted (Fig. 3c). It is worth noting that the EC film without Au NRs requires a certain voltage to achieve a similar level of modulation as the SLE4 film (Supplementary Fig. 13). At a lower voltage, the EC film makes more near-infrared light pass through and ensure a sufficient amount of visible light in the colored state (EC colored-1). At a higher voltage, the EC film absorbs a portion of visible light in the other colored state (EC colored-2) to achieve the absorption level of SLE4 film for NIR-1 (Supplementary Table 7). Thus, the selective absorption function of the film could continuously block near-infrared light without activating the electrochromic part, which meets the daylighting requirements and avoids the overreliance on modulation performance on electrochromic function.

Furthermore, the SLE4 film was encapsulated into a solid smart window device (Fig. 3d), with the help of H$_3$PO$_4$/polyvinyl alcohol (PVA) gel electrolyte[30]. When the solid smart window was installed on the model chamber, the solid SLE4 smart window can reduce blackbody temperature by 3.7 °C in bleached state and 4.8 °C in colored state (Fig. 3e and Supplementary Fig. 14). The simulated illumination direction was changed in the 30-degree intervals from 0° to 180° to simulate the change of actual sunlight irradiation angle, with each

position irradiated for half an hour. As shown in Fig. 3f, when the direction is perpendicular to the window for 0.5 h, the temperatures of outdoor air, blackbody in the EC window and SLE4 window chamber reach 34 °C, 33.0 °C, and 30.7 °C, respectively. The SLE4 solid window was powered by a standard 1.5 V battery when the irradiation direction was at 60°, and the peak temperature of the blackbody was only 28.9 °C. Unlike ordinary electrochromic smart windows, SLE windows do not require a long-time power supply for a better cooling effect. The Au NRs are regarded as a passive blocking component to selectively absorb near-infrared radiation, and the colored electrochromic W$_{18}$O$_{49}$ NWs serve as active blocking components to block more solar radiation when the sunlight becomes intense.

The switching time of electrochromic windows indicates whether the window can respond quickly in real-world application[31,32]. Figure 3g shows the switching time of the SLE4 film, and the 613.5 and 900 nm were selected as the representative wavelength of visible light and NIR-1, respectively. The vertical crossing network structure facilitates rapid electron transfer during the coloration process, enabling the SLE4 film to uniformly color or bleach in a matter of seconds. With a step voltage of −1.0 V and +0.2 V for 30 s, the SLE4 film can be colored in 4.5 s at 613.5 nm and 6.5 s at 900 nm, bleached in 4.0 s at 613.5 nm and 5.0 s at 900 nm. Coloration efficiency (CE) as another important indicator was studied as shown in Supplementary Fig. 15. The CE of SLE4 film was 48.46 cm$^2$ C$^{-1}$ at 613.5 nm and 86.23 cm$^2$ C$^{-1}$ at 900 nm. The transmittance difference ($\Delta T$) can be expressed as[33]:

$$\Delta T = \frac{T_{bleached,300}/T_{colored,300}}{T_{bleached,0}/T_{colored,0}} \quad (1)$$

By calculation, the $\Delta T$ remains about 96% at 613.5 nm and 93% at 900 nm after 300 switching cycles (Supplementary Fig. 16). The SLE4 film was further transferred on the flexible PC substrate to study its flexibility, which might be applied to curved surfaces of buildings. After 1000 bending cycles with a bending radius ($R$) of 1.5 cm, the $\Delta T$ of the

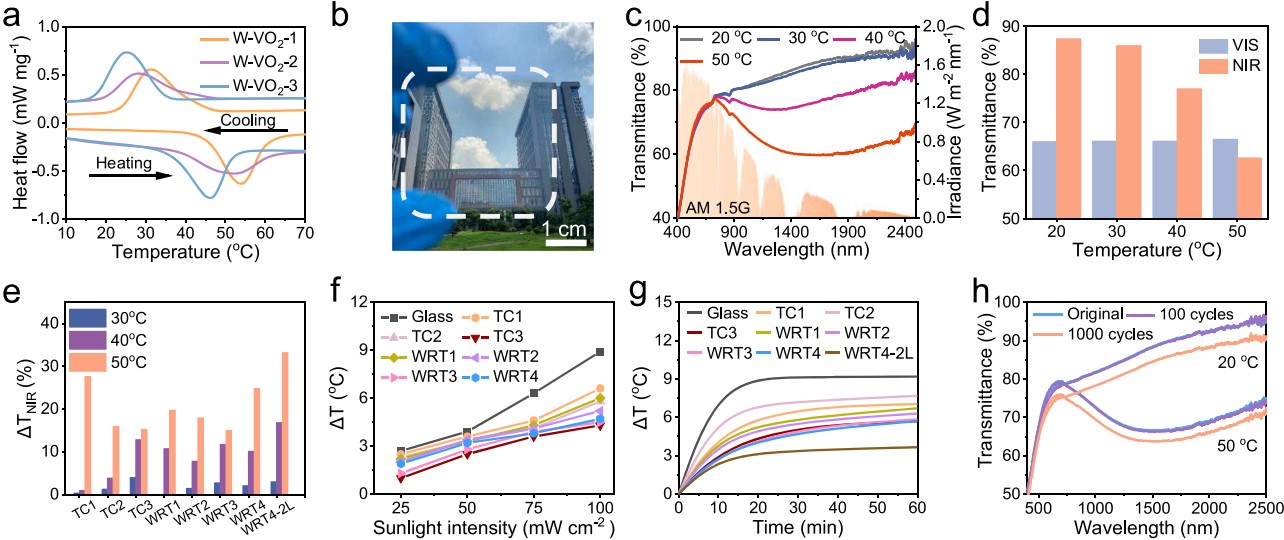

**Fig. 4 | Wide temperature stimulus-response range of smart windows. a** The differential scanning calorimetry curves for W-VO₂ NWs with different phase change temperature. **b** Photographs of WRT4 film. **c d** The transmittance spectra and the transmitted solar irradiance of WRT4 film at different temperatures under solar spectral irradiance. **e** The near-infrared light modulation ability of TC and WRT films at different temperatures. **f** The temperature change for blackbody in the model chamber installed with glass, TC and WRT films under the 100 mW cm⁻² simulated sunlight for 1 h. **g** The temperature change of blackbody in the model chamber installed with glass, TC and WRT films under the simulated illumination of different intensities. **h** The thermochromic cyclic stability of WRT4 film. Source data are provided as a Source Data file.

flexible SLE4 film in length (*L*) of 7 cm was only 2.3% under the same voltage (Fig. 3h). Moreover, due to the stability of the Au NRs and the overall device, the cooling performance of the SLE4 film shows almost no degradation even exposed to the real environment for up to 60 days (Supplementary Fig. 17). These results indicate the SLE film is a promising candidate for blocking solar radiation and daytime maintenance of appropriate indoor temperature in the decarbonization of buildings.

**Dynamic modulation of WRT smart windows**
In real weather conditions, the indoor demand for solar radiation varies due to dynamic changes in ambient temperature. Thus, the thermochromic smart window is regarded as a promising candidate for smart windows[34,35], the modulation of thermochromic materials can be realized spontaneously due to the phase transition triggered by changes in ambient temperature. Previously, element-doped VO₂ with lower phase change temperature has been used as effective thermochromic material in blocking solar radiation in hot weather, however, it will affect the building's demand for sufficient solar radiation in cold weather[36,37]. To solve this dilemma, the thermochromic smart windows with a wide stimulus-response range were fabricated via the same co-assembly strategy.

Firstly, the VO₂ NWs with various extents of W doping were synthesized by a hydrothermal method (Supplementary Figs. 18, 19), as observed in differential scanning calorimetry curves (Fig. 4a), the $T_c$ of W-VO₂-1, W-VO₂-2, and W-VO₂-3 NWs are 45, 40, and 35 °C, respectively. Then, the WRT films with good transparency (Fig. 4b and Supplementary Fig. 20) were fabricated by co-assembling the above three types of W-VO₂ NWs (Supplementary Table 8). The WRT films exhibit a wider temperature stimulus-response range when compared with thermochromic films (TC films) containing only one type of nanowires (Supplementary Fig. 21). Thus, the blocking effect of WRT films gradually changes with temperature instead of responding only in a narrow temperature range. Figure 4c, d show nearly 90% of the near-infrared light can pass through the WRT4 film at low temperature and less than 65% at high temperature, while the visible light transmittance of WRT4 film can maintain at about 65%, and the film has a little effect on the indoor daylighting no matter how the temperature changes. The ordered arrangement of NWs (Supplementary Fig. 22) effectively

reduces the haze of the films (10.9%), especially for the high haze (41.6%) caused by disordered W-VO₂ NWs with large diameters (Supplementary Fig. 23). Moreover, the films with 2 layers of mutually perpendicular nanowires (WRT4-2L) can even block more than 40% of near-infrared light at high temperature (Supplementary Fig. 24).

The thermochromic smart windows with a wide temperature stimulus-response range can achieve gradual modulation of sunlight, as the types and amounts of W-VO₂ NWs undergoing phase transition are different across a broad temperature range (Fig. 4e). We have calculated the optical transmittance and solar irradiance transmittance of TC and WRT films based on their spectral changes at different temperature (Supplementary Tables 9, 10). Smart windows with a wide temperature stimulus-response range are more suitable for the actual temperature change than thermochromic windows with a sharp and large transition. Specifically, the WRT4 window does not block sunlight when the temperature is below 30 °C. A certain number of nanowires undergo a phase transition between 30 and 40 °C, and the blocking ability is further enhanced as the temperature rises to 50 °C. This is also reflected in the simulated illumination results of different intensities (Fig. 4f). At low light intensity (25 and 50 mW cm⁻²), the cooling performance of WRT4 is close to that of TC1 and TC2 films with high phase change temperature, which means that more sunlight will enter the room through the WRT4 on cold days. At high light intensity (75 and 100 mW cm⁻²), the cooling effect of WRT4 is close to that of TC3 film with low phase change temperature, which indicates that more sunlight will be blocked in hot weather.

In Fig. 4g, the TC3, WRT3, and WRT4 films have similar cooling capabilities under 1 sun illumination, which reduces the temperature by about 3.5 °C compared to normal glass (Supplementary Figs. 25, 26). The reason for similar results is that the temperature of these films is below 40 °C under 1-sun illumination, and mainly W-VO₂-3 NWs in the films undergo the phase transition. When the illumination intensity was further increased to 1.5 (150 mW cm⁻²) and 2 (200 mW cm⁻²) times, the temperature of the film rises to about 45 and 48 °C (Supplementary Fig. 27). The temperature of the films increases as the light intensity increases, resulting in three kinds of W-VO₂ NWs undergoing a phase transition, thereby the blocking performance of the films is enhanced (Supplementary Figs. 28, 29). Dynamic regulation of WRT4 film

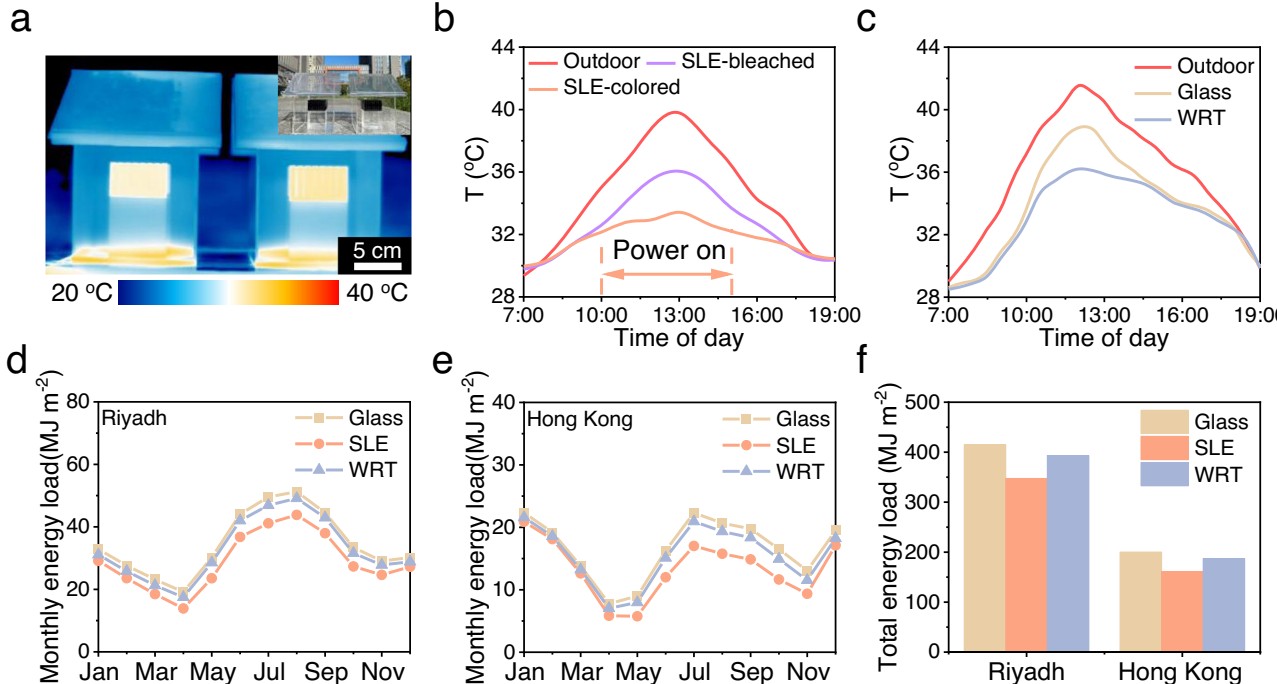

**Fig. 5 | Actual cooling performance and energy-saving simulation of smart windows. a** The infrared images of model houses installed with SLE and WRT smart windows, the insert is the photograph of model houses installed with SLE and WRT smart windows. **b**, **c** Measured temperature variation of indoor blackbody and outdoor in the SLE smart window house and WRT smart window house in September, Hefei. The windows are powered on from 10:00 (10 a.m.) to 15:00 (3 p.m.) to activate the electrochromic function. **d** Monthly energy load of the normal glass, SLE and WRT smart windows in Riyadh. **e** Monthly energy load of the normal glass, SLE and WRT smart windows in Hong Kong. **f** Calculated total energy load per year for the windows applied in Riyadh and Hong Kong, respectively. Source data are provided as a Source Data file.

performance with light intensity and temperature implies that the film as a thermochromic smart window adapts well to the spatially and temporally variable weather. After 1000 cycles, the film still exhibits good phase transition stability, and the transmittance at 2500 nm drops less than 5% (Fig. 4h). While the cooling performance of the WRT4 film has diminished by 0.6 °C at the same time, this is because the thermochromic properties of the W-VO$_2$ NWs are weakened by partial oxidation in the air (Supplementary Fig. 30).

### Energy saving efficiency of smart windows with large-area

Due to the lower production cost (Supplementary Tables 11 and 12) and large-scale preparation method, the large-area SLE and WRT films with the dimension of $25 \times 20$ cm$^2$ were prepared (Supplementary Fig. 31). To verify the effectiveness of smart windows at real ambient conditions, the identical house models installed with glass, SLE and WRT smart windows (Fig. 5a) were exposed to actual sunlight in Hefei, China (31°49′21″ N, 117°13′18″ E; 37.15 m altitude) in September. As shown in Fig. 5b, the temperature of outdoor ground rises from 29 to 39.8 °C. The blackbody in the model houses with these smart windows was recorded in real-time, due to the sustained absorption of near-infrared light by Au NRs, the peak temperature of the blackbody reaches 36 °C under the guarantee of the SLE windows. And the temperature was further reduced to 33.4 °C when the electrochromic function of the SLE windows was activated by an ordinary battery of 1.5 V at 10:00 (10 a.m.). In the model houses with WRT window, with the enhancement of solar radiation, more and more W-VO$_2$ NWs undergo phase transformation. Compared to the outdoor ground at 41 and 39 °C in the model house with normal glasses at noon, the blackbody temperature is only 36.1 °C with WRT windows. As the sunlight diminishes after 15:00 (3 p.m.), the blocking capacity of the WRT4 window dynamically decreased, resulting in the temperature in the model house with WRT4 windows and normal glasses tended to be consistent (Fig. 5c).

To further investigate the energy-saving performance of our smart windows, an energy-saving simulation was performed, which designs the actual-size building model in the simulation software (Supplementary Fig. 32). Climate data of Riyadh (Saudi Arabia) and Hong Kong (China) were selected to analyze the window performance. In the simulation, the electrochromic function of SLE window is activated when the temperature is above 30 °C, and the modulation of WRT window is dynamically adjusted with ambient temperature. Figure 5d, e describe the monthly total energy load of the normal glass, SLE, and WRT smart windows in Riyadh and Hong Kong. The monthly total energy load (including the HVAC system) of the building is significantly reduced by using these smart windows, especially in the hot summer months. Moreover, we calculated the energy saving by plotting the energy consumption difference between the smart windows and normal glass windows (Fig. 5f), In particular, the SLE window can reduce 16.3% and 19.6% total energy consumption in Riyadh and Hong Kong, respectively. As for the WRT window, 5.2% and 6.7% of total energy consumption decreased in Riyadh and Hong Kong, respectively (Supplementary Fig. 33). The simulation results indicate that the SLE and WRT smart windows show promising energy-saving performance in different cities. More, compared with the previous works, the electrochromic and thermochromic smart windows prepared by our LB co-assembly strategy show significant improvements in sunlight modulation and cooling effect, this simplicity and versatility of the co-assembly method allows for the large-scale preparation of smart windows (Supplementary Figs. 34, 35).

### Discussion

In summary, we propose a co-assembly strategy for the preparation of smart windows for solar regulation, and the optical performance of the windows can be significantly improved by modulating the components and structure of the multi-type materials. To tackle the bottleneck of

traditional electrochromic windows, Au NRs were introduced into the electrochromic window to selectively absorb near-infrared light without affecting visible light transmission as much as possible, and the absorption range can be broadened by co-assembling Au NR mixture with different aspect ratios. The electrochromic function of $W_{18}O_{49}$ NWs is used as an active option for further modulation of solar radiation according to the wishes of the households and the actual weather. The same multi-component co-assembly strategy can also improve the performance of conventional thermochromic windows, the thermochromic $W-VO_2$ NWs with different doping amounts are co-assembled to extend fixed response temperature value to a wide temperature range of 30–50 °C. The wide response range allows smart windows to progressively adjust their blocking ability as the temperature changes, dynamically regulating the room temperature in various real-world weather conditions. Besides, the ordered arrangement of nanowires effectively reduces the haze of smart windows and shows good visibility. The advances detailed in this work are achieved with a large-scale preparation method, which can be served as a testament to the commercial promise of smart windows based on electrochromic and thermochromic films for saving energy in buildings.

## Methods
### Synthesis of Au nanorods
All chemicals and solvents were purchased from Shanghai Chemical Reagent Co. Ltd. and used without further purification. The multiple sizes of Au NRs were prepared by a seeded-growth method[38]. The seed solution was prepared as follows: 0.5 mM $HAuCl_4$ (99.0%) and 0.2 M cetyltrimethyl ammonium bromide (CTAB, 99.0%) were dissolved in 10 mL deionized water. Immediately, 0.6 mL of fresh 0.01 M $NaBH_4$ (98.0%) and 0.4 mL deionized water were added to the mixture together with vigorous stirring until the mixture changed to brownish-yellow. Then, the seed solution was aged in dark. The growth solution was prepared by the following methods: 7.0 g of CTAB and 1.234 g of NaOL (98.0%) were dissolved in 250 mL deionized water, then 24 mL of 4 mM $AgNO_3$ (99.8%) solution was added, and keeping undisturbed at 30 °C for 15 min. 250 mL of 1 mM $HAuCl_4$ solution was added to the mixture under slow stirring until the solution became colorless, and a certain volume of HCl (36.0–38.0%) was introduced to adjust the pH. 1.25 mL of fresh 0.064 M ascorbic acid (99.7%) solution was added and vigorously stirred for 30 s. Finally, a small amount of seed solution was injected into the growth solution quickly. The resultant mixture continues to be stirred for 30 s and was placed at 30 °C. After at least 12 h of growth, the final products were isolated by centrifugation at 3500g for 30 min followed by the removal of the supernatant. No size and/or shape-selective fractionation was performed (the concentration of Au NRs was 0.040 mg mL$^{-1}$).

### Synthesis of $W_{18}O_{49}$ nanowires
Uniform $W_{18}O_{49}$ NWs with an average diameter of 5 nm and length of several micrometers were prepared by a modified solvothermal method[39]. 0.0001 g of poly(vinylpyrrolidone) (PVP, K-30, 99.8%) and 0.03 g of $WCl_6$ (99.9%) were dissolved in 40 mL of ethanol (99.7%) with vigorous stirring. The yellow solution was then transferred into a 50 mL Teflon-lined stainless-steel autoclave and placed at 180 °C for 24 h. After that, the final products were isolated by centrifugation at 3500 x g for 5 min and re-dispersed into 4 mL of ethanol for co-assembly (the concentration of $W_{18}O_{49}$ NWs was 0.004 g mL$^{-1}$).

### Synthesis of Ag nanowires
Ag NWs with an average diameter of 60 nm and length of several tens of micrometers were prepared by the polyol method[40]. 5.86 g of PVP (K-30) and 190 mL of glycerol (99%) were kept at 110 °C for about 1 h under vigorous stirring to form a homogeneous solution. 1.58 g of $AgNO_3$ (99.8%) was added to the solution after cooling down to 30 °C. Then the solution was heated to 210 °C for 20 min with slow stirring.

The NaCl solution (0.059 g of NaCl (99.5%) dissolved in 0.5 mL of deionized water and 10 mL of glycerol) was added to the solution at 50 °C. When the temperature reached 210 °C, the heating was stopped immediately and 200 mL of deionized water was added. The solution was kept undisturbed for 12 h to remove the Ag nanoparticles. The obtained Ag NWs were washed with ethanol to removal of the supernatant and re-dispersed into 100 mL of ethanol for co-assembly (the concentration of Ag NWs was 0.0084 g mL$^{-1}$).

### Surface treatment of Au nanorods
The CTAB on the surface of Au NRs was replaced by PVP (K-30, 99.8%) via ligand exchange[41], and the 25 mL of the as-prepared Au NRs was centrifuged and re-dispersed into 25 mL of 10 wt% PVP aqueous solution for overnight magnetic stirring. Then, the PVP-stabilized Au NRs were centrifuged at 3500g for 30 min to remove excess PVP and concentrated in 5 mL of N,N′-dimethylformamide (DMF, 99.5%) for co-assembly.

### Synthesis of $W-VO_2$ nanowires
$W-VO_2$ NWs with an average diameter of 200 nm and length of several tens of micrometers were prepared by a modified solvothermal method[42]. 1 g of $VOSO_4$ (99.9%) was dissolved in 40 mL of deionized water with vigorous stirring, and different amounts of $(NH_4)_6H_2W_{12}O_{40}\cdot xH_2O$ (99.5%) were introduced into the solution. 0.35 mL $N_2H_4$ (85 wt% water solution) was following added into the clear blue solution dropwise until the solution became gray, then continued to stir for 10 min. After that, the pH value of the solution was adjusted to 7 by 0.1 M NaOH (96%) solution. After suction filtration and centrifugal washing, the precipitate was dispersed in 40 ml of deionized water and transferred into a 100 mL Teflon-lined stainless-steel autoclave and placed at 250 °C for 36 h. After that, the final products were isolated by centrifugation at 1000g for 5 min and re-dispersed into 40 mL of ethanol for co-assembly (the concentration of $W-VO_2$ NWs was 0.009 g mL$^{-1}$).

### Fabrication of SLE and WRT films
The SLE and WRT films were fabricated by a modified LB co-assembly technique[43]. The co-assembly process was operated in the LB trough (Nima Technology, 312D) at room temperature using Millipore Milli-Q water (resistivity 18.2 MΩ.cm) as subphase. To obtain SLE films, 0.75 mL of $W_{18}O_{49}$ NWs solution and 0.6 mL of Ag NW solution were centrifuged and mixed into a solution of 1.0 mL DMF (99.5%), then 1.0 mL of $CHCl_3$ (99%) was added to the NWs mixture with sonicating. After that, the mixed NWs suspension was added onto the water subphase drop by drop with a syringe. The NWs were compressed with a compression rate of 20 cm$^2$ min$^{-1}$ until the formation of the fold that parallels the barrier direction. Different layers of the aligned $W_{18}O_{49}$/Ag NWs with crossing angles were transferred to the transparent polycarbonate or normal glass. After that, the multiple sizes of Au NRs were mixed for a total volume of 1 mL and 0.75 mL of $W_{18}O_{49}$ NW solution, which was centrifuged and dispersed into 1.0 mL DMF solution. Then 1.0 mL of $CHCl_3$ has added to the NRs and NWs mixture with sonicating. With similar interface assembly conditions, the obtained $W_{18}O_{49}$/Au layers were deposited on the $W_{18}O_{49}$/Ag NW layers. To obtain the WRT films, the $W-VO_2$ NWs with different phase transition temperatures were mixed for a total volume of 1 mL solution, which was centrifuged and dispersed into 1.0 mL DMF solution, then 1.0 mL of $CHCl_3$ was added to the NWs mixture with sonicating. The different layers of the aligned $W-VO_2$ NWs with crossing angles were transferred to the transparent polycarbonate substrate or normal glass.

### Measurement of the indoor cooling performance of the smart windows
The model chamber was built using recycled carton external structure brackets and wrapped with aluminum foil for thermal insulation on the

outside the dimensions of the model chamber are $15 \times 10 \times 10$ cm$^3$. The indoor cooling tests were done at room temperature. The model chamber was installed with a 10 cm$^2$ window. A black anodized cuboid was placed in the center of the chamber as a blackbody heat absorber. The window was exposed to continuous incident solar radiation simulated from a Xenon lamp (PLS-SXE3000, Perfect Light). The temperature versus time for different films was plotted, and the steady temperatures for the blackbody and air were compared. Infrared images and surface temperatures of the window samples were captured and recorded by a thermal infrared camera (VarioCAM®hr head 680, InfraTec).

### Calculations of Integrated optical and solar irradiance transmittance

Integrated optical transmittance ($T$) and solar irradiance transmittance ($T'$) in the VIS (400–760 nm), NIR-1 (760–1360 nm), NIR-2 (1360–2500 nm), NIR (760–2500 nm) regions, and entire solar (400–2500 nm) regions. the calculations were based on these equations:

$$T_{sol} = \frac{\int T(\lambda)d\lambda}{\int d\lambda} \qquad (2)$$

$$T_{sol'} = \frac{\int T(\lambda)\Psi(\lambda)d\lambda}{\int \Psi(\lambda)d\lambda} \qquad (3)$$

where $T(\lambda)$ is the transmittance at a wavelength of $\lambda$, and $\Psi(\lambda)$ is the solar irradiance spectrum for air mass 1.5 under the sun standing 37° above the horizon.

### Commercial building energy consumption simulations

The software of EnergyPlus was used to evaluate the energy consumption with different windows in different climate zones. An $8 \times 6 \times 2.7$ m$^3$ model house was built with two $5 \times 2$ m$^2$ windows and two $3 \times 2$ m$^2$ windows on four walls. The climate data of Riyadh and Hong Kong were selected to analyze the window performance in different climate types. Dual set points of 20 °C and 26 °C were applied for the heating and cooling of the HVAC system, respectively. The electrochromic part of ETW window is activated when the temperature exceeds 30 °C, and the thermochromic function of DTW window is dynamically adjusted with ambient temperature.

### Sample characterization

Transmission electron microscope (TEM) images were acquired using a Hitachi H-7700 transmission electron microscope operated at an accelerating voltage of 100 kV. Scanning electron microscopy (SEM) images were acquired by a field emission scanning electron microanalyzer (Zeiss Supra 40 scanning electron microscope at an acceleration voltage of 5 kV). The X-ray diffraction patterns (XRD) were measured on a Philips X' Pert Pro Super X-ray diffractometer equipped with graphite-monochromatized Cu KR radiation. The differential scanning calorimetry (DSC) curves were recorded on DSC Q2000 (TA Instruments). UV–VIS–NIR spectra were recorded on SolidSpec-3700 (Shimadzu Corporation) in transmittance mode with an integrating sphere at room temperature and equipped with TS120 heating & cooling stage (Wuhan Congtical Technology Corporation). The haze spectra were measured by a CS-700 Haze Meter (Hangzhou CHNSpec Technology) with a wavelength range of 400–760 nm. ICP-AES data were obtained by an Optima 7300 DV instrument (PerkinElmer Corporation). The sheet resistance of the film was tested by M-3 portable four-point probe test meter (Suzhou lattice electron Limited company). The mechanical stability of the devices was tested by a mechanical system (Instron 5565A). All the electrochemical measurements were carried out using a three-electrode configuration with a Pt wire as the counter electrode and Ag/AgCl as the reference electrode on a CHI760D electrochemistry workstation without separate instructions. The time-dependent infrared images were recorded by Thermal imager VarioCAM hr head 680 (InfraTec GmbH).

### Reporting summary

Further information on research design is available in the Nature Portfolio Reporting Summary linked to this article.

## Data availability

All the data supporting the findings of this study are available form the corresponding authors upon request. Source data are provided as a Source Data file. Source data are provided with this paper.

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

## Acknowledgements

The authors acknowledge the funding support from the National Natural Science Foundation of China (Grants 22175164, J.-W.L., 22005286, J.-L.W., and U1932213, 51732011, S.-H.Y.), the National Key Research and Development Program of China (Grant 2018YFE0202201, 2021YFA0715700, S.-H.Y.), Science and Technology Major Project of Anhui Province (Grant 201903a05020003, S.-H.Y.), and the University Synergy Innovation Program of Anhui Province (Grant GXXT-2019-028, S.-H.Y. and GXXT-2020-072, J.-W.L.). This work was partially carried out at the USTC Center for Micro and Nanoscale Research and Fabrication. This work has been supported by New Cornerstone Science Foundation.

## Author contributions

J.-W.L., S.-Z.S., and S.-H.Y. conceived the idea and designed the experiments. S.-H.Y., and J.-W.L. supervised the research. S.-Z.S., and J.-L.W. carried out the synthesis experiment, co-assembly experiment, and analysis. B.Z. and G.P. helped the energy-saving simulation. X.-F.F., and Q.-G.S., and J.Z. helped the UV–Vis–NIR spectra. Z.H. helped the TEM and SEM testing. C.C. helped the mechanical bending testing. S.-H.Y., J.-W.L., S.-Z.S., and J.-L.W. analyzed the data and co-wrote the manuscript. All authors analyzed and discussed the results.

## Competing interests

The authors declare no competing interests.
