## [Peer Review File · Nature Communications]

Nanowire-based smart windows combining electro- and thermochromic for dynamic regulation of solar radiationReviewers' Comments:

Reviewer #1:

Remarks to the Author:

In this manuscript, the authors fabricated composite photo-cum-electrochromic smart windows, with a fast coloration rate, which are capable of changing (lowering) room temperature up to 5 °C. While the strategy utilized to combine photochromic, electrochromic, and thermochromic layers to utilize both visible and NIR radiations for the color change of the composite films efficiently, apparently, the process of fabrication of the composite films is too complicated. I wonder whether the process can be followed by other researchers to reproduce the results easily.

On the other hand, the current version of the manuscript is a bit difficult to follow, probably due to the use of inappropriate expressions and phrases. For example, the sentences "The visible light region used for indoor lighting is very necessary but often overlooked, which is sacrificed for higher solar radiation modulation efficiency", and "Tiny amounts of Au nanorod assemblies with multiple aspect ratios are able to selectively block over 50% of near-infrared light radiation and maintain 70% visible light transmittance, with the help of W18O49 dynamically controlled by external voltages could effectively reduce the indoor temperature about 5 °C under 1-sun irradiation" are difficult to follow. On the novelty issue, several strategies for fabricating photochromic smart windows based on plasmonic and non-plasmonic nanomaterials have been reported recently (e.g., *Advanced Optical Materials*, <https://doi.org/10.1002/adom.202202171>; references 35, 26, 40, 42, etc.), which are simpler in structure and easier to fabricate.

In tables 8 & 9 (Section 17) of the supplementary Information, the authors presented the cost of production of the SLE4, and TCM4 films. While it is a bit confusing what they referred to by "fabrication costs of the materials in the unit area of SLE4 film", I suppose the labor cost for fabricating the films is not considered.

Reviewer #2:

Remarks to the Author:

The paper entitled "Nanowire-Based Smart Windows for Solar Radiation Regulation" shows solar radiation regulation by using plasmonic Au nanorods, electrochromic tungsten oxide nanowires and thermochromic Vanadium oxide nanowires. The authors propose new ideas of co-assembling different optical materials to have a high performance of smart window.

This manuscript shows detail characterizations of optical materials for smart window applications. There are some suggestions to improve the quality of the paper.

1. The authors suggest importance of modulation of sunlight at special band ranges. It is very good to have a detail description or demonstration of the modulation for some specific applications. The reader may not understand the point of this paper without proper demonstrations.
2. Please show reliability tests of the devices at real ambient conditions.

Reviewer #3:

Remarks to the Author:

The authors presented a generalized strategy to fabricate nanowire-based smart windows to modulate the solar heat gain with multispectral controllability. This work shows beautifully made nanowires with thorough characterization. The ordered structure produced by the Langmuir-Blodgett method is interesting and effective for improving the visual appearance (low haze). Still, I would request a few additional information and clarification before recommending for publication:

1. All of the nanowires used in this work were synthesized by following previous works, as noted in the manuscript's reference list. Moreover, their applications were the same – WO_x for electrochromics, W-VO₂ for thermochromics, AgNWs for transparent conductors, and AuNRs for plasmonic absorption.

After reading through the manuscript, I was still trying to grasp the novelty of this work, except for the L-B fabrication. For the readers of Nature Communications, it is essential to explain the novelty by providing the comparison chart with relevant prior works.

2. One potential advantage of W-VO₂ nanowires is to provide "wide-range thermochromics". While this seems to be a new function, I wonder what practical scenario would favor a gradual change versus a sharp one? The nanowires block only NIR light, so there isn't a concern about changing visual appearance. Shouldn't a sharp and large transition save more HVAC energy than a gradual one?

3. Based on the information provided in the current manuscript, it is difficult to interpret the temperature rise in the simulated house experiments. The optical measurement (transmittance) makes perfect sense because it is well-defined. However, the temperature rise is dependent not only on the window but the chamber and ambient convective heat transfer. How do we link a "2 degree Celsius drop" to cooling power or energy saving when it comes to HVAC thermal engineering?

Reviewer #4:

Remarks to the Author:

This manuscript has been focused on the fabrication of wavelength conversion material for solar light. Smart function of it has not been clarified based on the mechanism and basic principle on the oxidation state change of the ionic elements and the quantitative analysis of the transmittance and conversion efficiency were not supplied. Authors should define the basic principles of the wavelength conversion by the changing parameters of the W ion and quantitative analysis of the solar light wavelength change by determining transmittance at specific wavelength range. This is critical data for the wavelength conversion efficiency of the solar light regulation. So this manuscript should be rejected as the present form. Authors must add more required specific data to publish this manuscript.

The point-to-point answers to the Reviewers' questions:

We have carefully considered all valuable suggestions from the editor and five independent referees and have made suitable revisions accordingly. For clarity reasons, the revision part was marked in RED color and started with “**”.

Reviewers' comments:

Reviewer #1 (Remarks to the Author):

In this manuscript, the authors fabricated composite photo-cum-electrochromic smart windows, with a fast coloration rate, which are capable of changing (lowering) room temperature up to 5 °C. While the strategy utilized to combine photochromic, electrochromic, and thermochromic layers to utilize both visible and NIR radiations for the color change of the composite films efficiently, apparently, the process of fabrication of the composite films is too complicated. I wonder whether the process can be followed by other researchers to reproduce the results easily.

****Thank you for the kind comments. We apologize for the misunderstanding caused to the reviewer by our inaccurate descriptions and immature writing. In the revised version, we have rewritten the description of the preparation process and regulation mechanism of the devices. The novel electrochromic and thermochromic smart windows with tunable components and ordered structures are prepared by a co-assembly strategy, respectively.**

In Figure R1, the different aspect ratios of Au NRs with narrow absorption range are synthesized and mixed to cover the 760-1360 nm band, then the multi-sized Au NR mixture, electrochromic W₁₈O₄₉ NWs, and conductive Ag NWs are co-assembled into an ordered network-like structure as selective light absorption electrochromic (SLE) smart window. This window could selectively absorb near-infrared light and avoid the overreliance on electrochromic functions. When the negative voltage is applied, the device transitions to a colored state that can effectively block the majority of near-infrared light.

In Figure R2, the thermochromic windows based on one type of W-VO₂ NWs have a narrow temperature response range and are not suitable for actual weather changes. Compared with the fixed phase change temperature (68 °C) of VO₂, the thermochromic W-VO₂ NWs with different tungsten (W) element doping are co-assembled to extend fixed response temperature value to a wide temperature range of 30~50 °C. At low temperature, only the nanowires with high doping amounts undergo a phase change, while several nanowires with different doping amounts are in the phase transition state with the increase in temperature. This wide response range allows smart windows to progressively adjust their blocking ability as the temperature changes.

In this strategy, the components and structure of the device can be tuned by changing the mixing ratio of the precursor solution and the stacking angle of each layer of the film without complex material surface treatment and external field induction. Moreover, the simplicity of the co-assembly method enables the easy fabrication of large-area smart windows with dimensions of 25 × 20 cm², highlighting the potential for scalable production and promising real-world applications in diverse weather conditions.

Figure R1. The schematic illustration for the fabrication and modulation mechanism of electrochromic film with selective light absorption.

Figure R2. The schematic illustration for the fabrication and modulation mechanism of thermochromic film with a wide response-range.

On the other hand, the current version of the manuscript is a bit difficult to follow, probably due to the use of inappropriate expressions and phrases. For example, the sentences "The visible light region used for indoor lighting is very necessary but often overlooked, which is sacrificed for higher solar radiation modulation efficiency", and "Tiny amounts of Au nanorod assemblies with multiple aspect ratios are able to selectively block over 50% of near-infrared light radiation and maintain 70% visible light transmittance, with the help of $W_{18}O_{49}$ dynamically controlled by external voltages could effectively reduce the indoor temperature about 5 °C under 1-sun irradiation" are difficult to follow.

****Thank you for the kind comments. We have corrected the inappropriate expressions and phrases in the manuscript as follows:**

1) On page 2, line 3. "Here we report a new strategy for excellent solar radiation regulation of smart windows by co-assembling various optical materials into ordered structures, such as plasmonic Au nanorods, electrochromic $W_{18}O_{49}$ nanowires, and thermochromic VO_2 nanowires." **has been corrected to "Herein, electrochromic and thermochromic smart windows with tunable components and ordered structures are prepared by a co-assembly strategy to modulate sunlight and effectively manage heat, respectively."**

2) On page 2, line 5. "Tiny amounts of Au nanorod assemblies with multiple aspect ratios are able to selectively block over 50% of near-infrared light radiation and maintain 70% visible light transmittance, with the help of $W_{18}O_{49}$ dynamically controlled by external voltages could effectively reduce the indoor temperature about 5 °C under 1-sun irradiation." **has been corrected to "Firstly, to enhance both**

illumination and cooling efficiency in electrochromic windows, the aspect ratio and mixed type of Au nanorods are tuned to selectively absorb near-infrared light range of 760-1360 nm. Furthermore, when assembled with electrochromic $W_{18}O_{49}$ nanowires in colored state, the Au nanorods exhibit a synergistic effect, resulting in a remarkable 90% reduction of near-infrared light and a corresponding 5 °C cooling effect under 1-sun irradiation.”

3) On page 2, line 10. “Similarly, tungsten-doped VO_2 nanowire assemblies with different phase transition temperatures are introduced to spontaneously tune modulation performance under ambient temperature variations, reducing the energy consumption for device operation.” has been corrected to “Secondly, to extend the fixed response temperature value to a wider range of 30-50 °C in thermochromic windows, the doping amount and mixed type of W- VO_2 nanowires are carefully regulated.”

4) On page 2, line 13. “Besides, the ordered arrangement of nanowires can significantly reduce the haze and bring high visibility.” has been corrected to “Last but not the least, the ordered assembly structure of the nanowires can greatly reduce the level of haze and enhance visibility in the windows.”

5) On page 2, line 15. “the energy consumption associated with improving the occupants’ comfort has accounted for 15% of the global energy consumption and will grow exponentially in the next few decades.” has been corrected to “the energy consumption required to improve occupants' comfort accounts for approximately 15-20% of global energy usage, a figure which is expected to increase exponentially in the coming decades.”

6) On page 2, line 17. “Especially, the excessive use of air conditioning to create a comfortable living temperature causes a large amount of power and greenhouse gas emissions” has been corrected to “In particular, the overreliance on air conditioning to achieve comfortable indoor temperatures results in significant power consumption and corresponding greenhouse gas emissions.”

7) On page 2, line 18. “Related research shows that 50% of energy consumption for cooling, heating, and lighting can be saved by installing windows with reversible optical properties in buildings to adjust the transmission of solar radiation.” has been corrected to “Research in this field has shown that installing windows with reversible optical properties can result in a 50% reduction in energy consumption for cooling, heating, and lighting in buildings”.

8) On page 2, line 20. “While the near-infrared light (NIR) region (760-2500 nm) accounts for about 50% of the solar radiation, which produces interior heating but has no effect on daylighting.” has been corrected to “The effectiveness of daylighting in buildings relies heavily on the visible light (VIS) spectrum, which ranges from 400 to 760 nm. The remaining 50% of solar radiation is primarily located in the near-infrared (NIR) region, ranging from 760 to 2500 nm, and is mostly utilized for interior heating purposes”.

9) On page 3, line 6 “As a viable technology to economize building energy consumption, smart windows based on electrochromic and thermochromic materials have been reported to regulate the transmission of sunlight by the reversible switching of optical properties under external stimulation.” has been corrected to “As a viable technology to economize building energy consumption, smart windows based on electrochromic and thermochromic materials have been widely reported due to the reversible optical transitions.”

10) On page 3, line 10. “Furthermore, various micro- and nano-structures are used for changing the color variation range of these materials, which allows these smart windows to meet different applications.” has been corrected to “Furthermore, various micro- and nano-structures are used for improving the performance of these materials to meet different applications.”

11) On page 3, line 13. “while ignoring the actual needs of buildings for specific bands of sunlight.” has been corrected to “without intelligent control over specific bands of sunlight that buildings require based on their individual needs.”

12) On page 3, line 14. “The visible light region used for indoor lighting is very necessary but often overlooked, which is sacrificed for higher solar radiation modulation efficiency.” has been corrected to “Higher visible region transmission is critical for indoor lighting but is often sacrificed for higher modulation efficiency.”

13) On page 3, line 19. “we propose a general and feasible co-assembly strategy to fabricate spectrally selective and dynamic smart windows by adjusting the components and assembly structure of materials.” has been corrected to “we propose a general and feasible co-assembly strategy to fabricate smart windows for solar regulation by adjusting the multi-components and assembly structure.”

14) On page 3, line 20. “The electrochromic $W_{18}O_{49}$ nanowires (NWs) were assembled with tiny amounts of Au nanorods (NRs) for efficient indoor temperature management.” has been corrected to “By integrating tiny amounts of plasmonic Au nanorods (NRs) with electrochromic $W_{18}O_{49}$ nanowires (NWs), the optical modulation performance of conventional electrochromic windows has been significantly improved.”

15) On page 4, line 1. “The Au NRs with different aspect ratios produce enhanced broadband absorption of near-infrared light to block more than 50% of near-infrared light.” has been corrected to “A mixture of Au NRs with varying aspect ratios exhibits strong absorption within the specific wavelength range of 760-1360 nm, allowing for the blocking of over 50% of light within this band.”

16) On page 4, line 3. “After being applied with a negative voltage of 1.5 V, the device in colored state actively blocks most of the near-infrared light (more than 90%) and reduces about 5 °C of the indoor temperature under 1 sun irradiation (the maximum natural sunlight).” has been corrected to “When a negative voltage of 1.5 V is applied, the device transitions to a colored state that can effectively block the majority of near-infrared light (over 90%), resulting in a temperature reduction of approximately 5 °C indoors under 1 sun irradiation (maximum natural sunlight).”

17) On page 4, line 7. “The same co-assembly strategy is applied to improve the dynamic response range of traditional thermochromic windows.” has been corrected to “This strategy of multi-component modulation in devices can also be applied to improve the performance of traditional thermochromic windows.”

18) On page 4, line 9. “Thermochromic VO_2 NWs with various extents of tungsten (W) doping were co-assembled in the same strategy to enhance the dynamic regulation of smart windows.” has been corrected to “Compared with the fixed phase change temperature (68 °C) of VO_2 , the thermochromic W- VO_2 NWs with different amount of tungsten (W) doping were co-assembled to broaden the temperature stimulus response of smart windows.”

19) On page 4, line 11. “The temperature stimulus-response is broadened to the range of 30 °C~50 °C, which means that the modulation performance of the windows can spontaneously change with ambient temperature without any electrical power.” has been corrected to “These smart windows, which have a wide response range of 30°C to 50°C, can dynamically adjust their blocking performance. As the ambient temperature increases, the windows' ability to block sunlight becomes stronger.”

20) On page 4, line 13. “Besides, due to the simple co-assembly method, large-area smart windows with the dimension of 25 × 20 cm² can be easily fabricated, demonstrating the possibility of the scale-up fabrication for promising applications in real weather conditions.” has been corrected to “Moreover, the simplicity of the co-assembly method enables the easy fabrication of large-area smart windows with dimensions of 25 × 20 cm², highlighting the potential for scalable production and promising real-world applications in diverse weather conditions.”

21) On page 4, line 17, “It has been reported that the performance of the devices can be enhanced by co-assembling multiple types of materials based on the interface co-assembly method. Herein, multiple components based smart windows are designed and fabricated based on the co-assemble method to

improve the actual performance of conventional tinted windows.” have been corrected to “Interface co-assembly method is a versatile and effective approach to prepare smart windows with tunable properties, which involves the simultaneous assembly of multiple nanomaterials at an interface to create a functional composite structure.”

22) On page 4, line 19, “In brief, the multi-sized Au NRs with strong light absorption properties, electrochromic $W_{18}O_{49}$ NWs, and conductive Ag NWs are co-assembled into an ordered network-like structure as selective light absorption electrochromic (SLE) smart windows to selectively block near-infrared light in Fig. 1a. ” has been corrected to “ Fig. 1a illustrated the schematic fabrication of our smart windows. To create a selective light absorption electrochromic (SLE) smart window, different aspect ratios of Au NRs with narrow absorption ranges are synthesized and mixed to cover the 760-1360 nm band. Then, the multi-sized Au NR mixture, along with electrochromic $W_{18}O_{49}$ NWs and conductive Ag NWs, are co-assembled into an ordered network-like structure for precise modulation of the NIR region. This window is capable of selectively absorbing near-infrared light from 760 to 1360 nm for both illumination and cooling efficiency, and the synergy of these optical materials can significantly improve the cooling effect at higher temperatures.”

23) On page 5, line 5. “Analogously, a variety of thermochromic VO_2 NWs with different W doping contents are also co-assembled into a similar structure as wide-range thermochromic (WRT) smart windows to optimize the dynamic modulation ability at different ambient temperatures.” has been corrected to “Using the same interfacial assembly regulation strategy, a wide response-range thermochromic (WRT) smart window was developed by co-assembling thermochromic $W-VO_2$ NWs with different amount of W doping, resulting in a wide response range of 30 °C to 50 °C, extending the fixed response temperature value. Compared to the narrow temperature response range of thermochromic windows using a single type of VO_2 NWs, the WRT smart window uses $W-VO_2$ NWs with different phase transition temperatures to achieve a wide response range of 30 °C to 50 °C. This is because only the nanowires with high doping amounts undergo a phase change at low temperature, while several nanowires with different doping amounts are in the phase transition state as the temperature increases. This broad response range enables smart windows to gradually modulate their light-blocking performance as the ambient temperature varies, providing effective control over indoor lighting and temperature (Fig. 1b).”

24) On page 5, line 14. “As shown in Fig. 1c, when the house is equipped with these smart windows, they will selectively block solar radiation and dynamically regulate indoor temperature under a small applied voltage or ambient temperature change.” have been corrected to “As shown in Fig. 1c, when the smart windows are installed in a house, they selectively block solar radiation and dynamically regulate the indoor temperature in response to a small applied voltage or ambient temperature change.”

25) On page 5, line 17. “The Au NRs of different sizes are co-assembled with electrochromic $W_{18}O_{49}$ NWs to selectively absorb the near-infrared light, especially 760-1360 nm (NIR-1), which accounts for three-quarters of the solar radiation in near-infrared light region. The Au NRs with localized surface plasmon resonance (LSPR) effect can only absorb light of specific wavelengths due to determined morphology.” have been corrected to “The Au NRs with localized surface plasmon resonance (LSPR) effect can only absorb light of specific wavelengths due to determined morphology. To achieve selective absorption of NIR light, especially the band that occupies three-quarters of the energy in the NIR region (NIR-1: 760-1360 nm).”

26) On page 6, line 5. “Ag NWs were used to replace traditional rigid ITO conductive glass as the conductive layer, and ultrafine electrochromic $W_{18}O_{49}$ NWs not only separated Au NRs and Ag NWs to ensure visible light transmittance but also further blocked solar radiation in the colored state.” has been corrected to “ The conductive layer was created using Ag NWs instead of traditional rigid ITO

conductive glass, while ultrafine electrochromic $W_{18}O_{49}$ NWs were utilized to separate Au NRs and Ag NWs, thereby ensuring visible light transmittance, and to further block solar radiation in the colored state.”

27) On page 6, line 10. “The resultant orderly cross-aligned structure can significantly reduce the haze of the films from 37.7% to 14% (Supplementary Fig. 6), which can be attributed to the disordered arrangement of NWs leading to more scattering of incident light.” has been corrected to “The resultant orderly cross-aligned structure can significantly decrease the scattering of light in all directions, and thus reduces the haze of the films from 37.7% of disordered structure to 14% (Supplementary Fig. 6).”

28) On page 6, line 16. “Taking the requirements of the optical property and sheet resistance into account, 6 layers of $W_{18}O_{49}$ NW networks and 4 layers of $W_{18}O_{49}/Ag$ NW networks were selected as the electrochromic (EC) film (Supplementary Fig. 8) for further study of electrochromic performance, while 6 layers of $W_{18}O_{49}/Au$ networks and 4 layers of $W_{18}O_{49}/Ag$ NW networks were selected as the selective light absorption electrochromic (SLE) film (Fig. 2b).” has been corrected to “Taking into account the requirements for optical properties and sheet resistance, we laminated 6 layers of $W_{18}O_{49}$ NW networks and 4 layers of $W_{18}O_{49}/Ag$ NW networks to form the electrochromic (EC) film. In contrast, 6 layers of $W_{18}O_{49}/Au$ networks and 4 layers of $W_{18}O_{49}/Ag$ NW networks were stacked to form the selective light absorption electrochromic (SLE) film (Fig. 2b).”

29) On page 7, line 3. “With the types of Au NRs increasing, the absorption range of the near-infrared range becomes wider, which covers the NIR-1 solar radiation very well (Fig. 2d).” has been corrected to “With the types of Au NRs increasing, the absorption range of the near-infrared range becomes wider to cover the entire NIR-1 region (Fig. 2d).”

30) On page 7, line 12. “a model chamber with a blackbody of $3.0 \times 3.0 \times 1.5 \text{ cm}^3$ as an absorber in it was used to study the temperature regulation performance of the fabricated EC and SLE films.” has been corrected to “to investigate the temperature regulation performance of the fabricated EC and SLE films, a model chamber with a $3.0 \times 3.0 \times 1.5 \text{ cm}^3$ blackbody absorber was utilized.”

31) On page 7, line 21. “SLE4 films can switch between colored and bleached states when applied with a voltage (versus $Ag/AgCl$) of -1.0 V or +0.2 V in 1M $LiClO_4$ /polycarbonate electrolyte solution (Supplementary Fig. 12).” has been corrected to “EC and SLE4 films can switch between colored and bleached states when applied with a voltage (versus $Ag/AgCl$) from -1.0 V to +0.2 V in 1M $LiClO_4$ /polycarbonate electrolyte solution (Supplementary Fig. 12).”

32) On page 8, line 14. “When the window is switched on, the temperature of the blackbody decreases from 28.6 °C to 27.5 °C (Fig. 3e and Supplementary Fig. 13). The solid SLE4 smart window can reduce 3.7 °C and 4.8 °C at different states, meaning that a lot of air conditioning energy consumption can be saved.” have been corrected to “When the solid smart window was installed on the model chamber, the solid SLE4 smart window can reduce blackbody temperature by 3.7 °C in bleached state and 4.8 °C in colored state (Fig. 3e and Supplementary Fig. 14).”

33) On page 8, line 16. “The simulated illumination direction was changed from 0° to 180° at 30° intervals to simulate the change of actual sunlight irradiation angle, and every position was irradiated for half an hour.” have been corrected to “The simulated illumination direction was changed in 30-degree intervals from 0° to 180° to simulate the change of actual sunlight irradiation angle, with each position irradiated for half an hour.”

34) On page 9, line 7. “Due to the vertical crossing network structure, electrons can be transferred quickly in the process of coloration, so that the SLE4 film can be uniformly colored or bleached in a few seconds.” has been corrected to “The vertical crossing network structure facilitates rapid electron transfer during the coloration process, enabling the SLE4 film to uniformly color or bleach in a matter of seconds.”

35) On page 10, line 5. “in which modulation systems without power input will be realized spontaneously due to the phase transition of thermochromic materials that can be triggered by the ambient temperature change.” has been corrected to “the modulation of thermochromic materials can be realized spontaneously due to the phase transition triggered by changes in ambient temperature.”

36) On page 10, line 9. “To solve this dilemma, the thermochromic smart windows with a dynamic stimulus-response range were fabricated via the same co-assembly strategy.” has been corrected to “To solve this dilemma, the thermochromic smart windows with a wide stimulus-response range were fabricated via the same co-assembly strategy.”

37) On page 10, line 14. “Then, the wide response-range thermochromic (WRT) films with good transparency (Fig. 4b and Supplementary Fig. 18) were fabricated by co-assembling the different W doped VO₂ NWs (Supplementary Table 7).” has been corrected to “Then, the wide response-range thermochromic (WRT) films with good transparency (Fig. 4b and Supplementary Fig. 20) were fabricated by co-assembling the above three types of W-VO₂ NWs (Supplementary Table 8).”

38) On page 10, line 16. “The WRT films have a wider temperature stimulus-response range when compared with thermochromic films (TC films) containing only one kind of W-VO₂ NWs, such as TC1, TC2, and TC3 films (Supplementary Fig. 19).” has been corrected to “The WRT films exhibit a wider temperature stimulus-response range when compared with thermochromic films (TC films) containing only one type of nanowires (Supplementary Fig. 21).”

39) On page 11, line 6. “In the WRT films, different amounts of W-VO₂ NWs were mixed to optimize the specific thermochromic property of the films. Fig. 4e shows the near-infrared light modulation ability (ΔT_{NIR}) of TC and WRT films at different temperatures, the higher the content of W-VO₂-3 NWs in the film is, the more suitable to block solar radiation at lower temperatures. Thus, the WRT1 film consisting of W-VO₂-1 and W-VO₂-2 NWs has a strong blocking ability above 40 °C, and WRT3 film is made up of W-VO₂-2 and W-VO₂-3 NWs shows strong blocking performance below 40 °C. The radiation blocking ability of WRT4 film containing three kinds of W-VO₂ NWs can dynamically modulate solar radiation in a wide temperature range of 30-50 °C.” have been corrected to “The thermochromic smart windows with a wide temperature stimulus-response range can achieve gradual modulation of sunlight, as the types and amounts of W-VO₂ nanowires undergoing phase transition are different across a broad temperature range (Fig. 4e). We have calculated the optical transmittance and solar irradiance transmittance of TC and WRT films based on their spectral changes at different temperature (Supplementary Table 8 and 9). Smart windows with a wide temperature stimulus-response range are more suitable for the actual temperature change than thermochromic windows with a sharp and large transition. Specifically, the WRT4 window does not block sunlight when the temperature is below 30 °C. A certain amount of nanowires undergo a phase transition within 30 °C~40 °C, and the blocking ability is further enhanced as the temperature rises to 50 °C. This is also reflected in the results of the simulated illumination of different intensities (Fig. 4f). At low light intensity (25 and 50 mW/cm²), the cooling performance of WRT4 is close to that of TC1 and TC2 films with high phase change temperature, which means that more sunlight will enter the room through the WRT4 on cold days. At high light intensity (75 and 100 mW/cm²), the cooling effect of WRT4 is close to that of TC3 film with low phase change temperature, which indicates that more sunlight will be blocked in hot weather.”

40) On page 11, line 21. “The reason for these results is that the temperature of the films themselves is lower than 40 °C under 1 sun illumination, and mainly W-VO₂-3 NWs in the films undergo the phase transition.” has been corrected to “The reason for similar results is that the temperature of these films is below 40°C under 1 sun illumination, and mainly W-VO₂-3 NWs in the films undergo the phase transition.”

41) On page 12, line 4. “The higher the illumination intensity is, the more W-VO₂ NWs undergo the

phase transition, thereby the blocking performance of the films is enhanced.” has been corrected to “The temperature of the films increases as the light intensity increases, resulting in three kinds of W-VO₂ NWs undergoing a phase transition, thereby the blocking performance of the films is enhanced (Supplementary Fig. 28 and 29).”

42) On page 12, line 6. “The above results show the adaptability of the WRT4 film as a thermochromic smart window, and the performance of the smart window changes dynamically with the actual temperature in different weather.” has been corrected to “Dynamic regulation of WRT4 film performance with light intensity and temperature implies that the film as a thermochromic smart window adapts well to the spatially and temporally variable weather.”

43) On page 12, line 12. “It should be noted, the costs of the noble metals Au NRs and Ag NWs in the unit area of the SLE window (Supplementary Table 8) are less than 5.23 and 0.079 dollars per square meter, respectively, the cost of W-VO₂ NWs is 0.294 dollars per square meter (Supplementary Table 9).” has been corrected to “Due to the lower production cost (Supplementary Tables 11 and 12) and large-scale preparation method, the large-area SLE and WRT films with the dimension of 25 × 20 cm² were prepared (Supplementary Fig. 31).”

44) On page 12, line 14. “After that, identical house models installed with SLE4 and WRT4 windows (Fig. 5d) were exposed to actual sunlight in Hefei, China (31°49'21" N, 117°13'18" E; 37.15 m altitude) in September.” has been corrected to “To verify the effectiveness of smart windows at real ambient conditions, the identical house models installed with glass, SLE and WRT smart windows (Fig. 5a) were exposed to actual sunlight in Hefei, China (31°49'21" N, 117°13'18" E; 37.15 m altitude) in September.”

45) On page 13, line 3. “After 15:00, the temperature in the model house with WRT4 windows and bare glasses tended to be consistent under the weakening of sunlight and the decrease of external temperature (Fig. 5f).” has been corrected to “As the sunlight diminishes after 15:00, the blocking capacity of the WRT4 window dynamically decreased, resulting in the temperature in the model house with WRT4 windows and bare glasses tended to be consistent (Fig. 5c).”

46) On page 14, line 1. “In summary, we fabricate spectrally selective and dynamic smart windows based on ordered one-dimension nanomaterials assemblies with unique optical properties.” has been corrected to “In summary, we propose a co-assembly strategy for the preparation of smart windows for solar regulation, and the optical performance of the device can be significantly improved by modulating the components and structure of the multi-type materials.”

47) On page 14, line 3. “A small amount of Au NRs assemblies with multiple sizes are introduced to extend the range of near-infrared light absorption and continuously block over 50% of near-infrared light radiation. When the sunlight intensity and ambient temperature increase, the electrochromic W₁₈O₄₉ NWs can further enhance the blocking of solar radiation from bleached to colored state, which could reduce the indoor temperature by 5 °C under the synergistic effect of Au NRs.” have been corrected to “To tackle the bottleneck of traditional electrochromic windows, Au NRs were introduced into the electrochromic window to selectively absorb near-infrared light without affecting visible light transmission as much as possible, and the absorption range can be broadened by co-assembling Au NR mixture with different aspect ratios. The electrochromic function of W₁₈O₄₉ NWs is used as an active option for further modulation of solar radiation according to the wishes of the households and the actual weather.”

48) On page 14, line 8. “Under the same co-assembly strategy, the W-VO₂ NWs assemblies with different phase transition temperatures are adjusted to enhance response over a wide temperature range of 30 °C~50 °C.” has been corrected to “The same multi-component co-assembly strategy can also improve the performance of conventional thermochromic windows, the thermochromic W-VO₂ NWs with different doping amounts are co-assembled to extend fixed response temperature value to a wide temperature range of 30 °C~50 °C.”

49) On page 14, line 10. “Thus, the higher the ambient temperature is, the more solar radiation smart windows block out.” has been corrected to “the wide response range allows smart windows to progressively adjust their blocking ability as the temperature changes, dynamically regulating the room temperature in various real-world weather conditions.”

On the novelty issue, several strategies for fabricating photochromic smart windows based on plasmonic and non-plasmonic nanomaterials have been reported recently (e.g., *Advanced Optical Materials*, <https://doi.org/10.1002/adom.202202171>; references 35, 26, 40, 42, etc.), which are simpler in structure and easier to fabricate.

** Thank you for the comments. We apologize for not accurately conveying our innovation to the reviewer through our inaccurate descriptions and immature writing. It is essential to prepare high-performance smart windows by a simple and efficient method.

In this excellent work mentioned by the reviewer (*Advanced Optical Materials*, <https://doi.org/10.1002/adom.202202171>), the researchers present a novel approach for fabricating photo-chromic smart windows with high coloration contrast and fast color switching, integrating plasmonic yolk-shell nano-phosphors (PYSNPs) at the surface of photochromic WO₃ layers. While this photochromic smart window needs to absorb part of the visible light to excite the fluorescent material resulting in reduced transmission of visible light, which will affect the lighting of the room.

In reference 35 (new reference 36), the researchers fabricate centimeter-scale-patterned TiN nanoparticle arrays by a combination of nanotemplate technique and direct nitridation of the titanium oxide nanoparticles. The TiN nanoparticles are coated with a pure monoclinic phase (M-phase) VO₂ thermochromic film prepared by annealing after physical vapor deposition. However, complex deposition equipment and strict conditions are required to prepare the samples in this method, which is not conducive to the preparation of large size samples and the tuning of specific properties.

In reference 26 (new reference 27), the researchers report a new route for large area co-assembly of nanowires, resulting in the formation of multilayer ordered nanowire networks with tunable conductivity (7-40 Ω/sq) and transmittance (58-86% at 550 nm) for fabrication of flexible transparent electrochromic devices. While this device is unable to selectively modulate the spectrum and absorbs too much visible light in the colored state for good modulation performance.

In references 40 (new reference 41), the researchers report the preparation of Au NRs/PNIPAM composite hydrogels with fast thermal/optical response, high heating rate, and high structural integrity by a traditional electrospinning technique. The size of the Au nanorods in the device is single, which leads to a narrow spectrally selective absorption range of the device, and the practical effect is not strong.

In reference 42 (new reference 43), the researchers demonstrate a novel multicolor electrochromic device by co-assembling W₁₈O₄₉ and V₂O₅ nanowires using the solution-based Langmuir-Blodgett technique. The authors pay attention to the discoloration of device in the visible region, ignoring the fact that near-infrared light is equally important in regulating room temperature.

In our work, interface co-assembly method is a versatile and effective approach to prepare smart windows with tunable properties, which involves the simultaneous assembly of multiple nanomaterials at an interface to create a functional composite structure. In the new version of the manuscript, we propose a co-assembly strategy for the preparation of smart windows for solar regulation. By integrating tiny amounts of plasmonic Au nanorods (NRs) with electrochromic W₁₈O₄₉ nanowires (NWs), the optical modulation performance of conventional electrochromic windows has been significantly improved. A

mixture of Au NRs with varying aspect ratios exhibits strong absorption within the specific wavelength range of 760-1360 nm, allowing for the blocking of over 50% of light within this band. When a negative voltage of 1.5 V is applied, the device transitions to a colored state that can effectively block the majority of near-infrared light (over 90%), resulting in a temperature reduction of approximately 5 °C indoors under 1 sun irradiation (maximum natural sunlight). Significantly, the ordered arrangement of these nanomaterials reduces the haze of windows while still maintaining visible light transmittance of 70%. This strategy of multi-component modulation in devices can also be applied to improve the performance of traditional thermochromic windows. Compared with the fixed phase change temperature (68 °C) of VO₂, the thermochromic W-VO₂ NWs with different amounts of tungsten (W) doping were co-assembled to broaden the temperature stimulus response of smart windows. These smart windows, which have a wide response range of 30°C to 50°C, can dynamically adjust their blocking performance. As the ambient temperature increases, the windows' ability to block sunlight becomes stronger. Moreover, the simplicity of the co-assembly method enables the easy fabrication of large-area smart windows with dimensions of 25 × 20 cm², highlighting the potential for scalable production and promising real-world applications in diverse weather conditions.

In **Figure R3 and R4 (Supplementary Figure 33 and 34)**, compared with the previous work, the electrochromic and thermochromic smart windows prepared by our LB co-assembly strategy show significant improvements in sunlight modulation and cooling effect. This simplicity and versatility of the co-assembly method allows for the large-scale preparation of smart windows.

Figure R3 (Supplementary Figure 33). **a**, Summary of electrochromic performance (transmittance modulation and switching time) in some of the best reported and selected works. **b**, The radar plot in which the transmittance modulation, switching time, coloration efficiency, cyclic stability, and scalability of Tungsten Oxide based thermochromic windows are compared (Nanostructure: Ref 1, Multilayer: Ref 9, Amorphous: Ref 11, Doping: Ref 16, Surface-decorated: Ref 20).

Reference

Nanostructure:

1. Gu, H. *et al.* Highly efficient, near-infrared and visible light modulated electrochromic devices based on polyoxometalates and W₁₈O₄₉ nanowires. *ACS Nano* **12**, 559-567 (2018).
2. Wang, J.-L., Lu, Y.-R., Li, H.-H., Liu, J.-W. & Yu, S.-H. Large area co-assembly of nanowires for flexible transparent smart windows. *J. Am. Chem. Soc.* **139**, 9921-9926 (2017).
3. Heo, S., Kim, J., Ong, G. K. & Milliron, D. J. Template-free mesoporous electrochromic films on flexible substrates from tungsten oxide nanorods. *Nano Lett.* **17**, 5756-5761 (2017).
4. Li, H., McRae, L. & Elezzabi, A. Y. Solution-processed interfacial PEDOT:PSS assembly into porous tungsten molybdenum oxide nanocomposite films for electrochromic applications. *ACS Appl. Mater. Interfaces* **10**, 10520-10527 (2018).

5. Nguyen, T. D. et al. Efficient near infrared modulation with high visible transparency using SnO₂-WO₃ nanostructure for advanced smart windows. *Adv. Optical Mater.* 7, 1801389 (2019).
6. Qu, H. et al. Highly robust and flexible WO₃·2H₂O/ PEDOT films for improved electrochromic performance in near-infrared region. *Sol. Energy Mater Sol. Cells* 163, 23-30 (2017).

Multilayer :

7. Li, H., Lv, Y., Zhang, X., Wang, X. & Liu, X. High-performance ITO-free electrochromic films based on bi-functional stacked WO₃/Ag/WO₃ structures. *Sol. Energy Mater Sol. Cells* 136, 86-91 (2015).
8. Xiao, L., Lv, Y., Dong, W., Zhang, N. & Liu, X. Dual-functional WO₃ nanocolumns with broadband antireflective and high-performance flexible electrochromic properties. *ACS Appl. Mater. Interfaces* 8, 27107-27114 (2016).
9. Najafi-Ashtiani, H., Akhavan, B., Jing, F. & Bilek, M. M. Transparent conductive dielectric-metal-dielectric structures for electrochromic applications fabricated by high-power impulse magnetron sputtering. *ACS Appl. Mater. Interfaces* 11, 14871-14881 (2019).
10. Dong, W. et al. Bifunctional MoO₃-WO₃/Ag /MoO₃-WO₃ films for efficient ITO-free electrochromic devices. *ACS Appl. Mater. Interfaces* 8, 33842-33847 (2016).

Amorphous:

11. Cheng, W. et al. Photodeposited amorphous oxide films for electrochromic windows. *Chem* 4, 821-832 (2018).
12. Huo, X. et al. Bifunctional aligned hexagonal/amorphous tungsten oxide core/shell nanorod arrays with enhanced electrochromic and pseudocapacitive performance. *J. Mater. Chem. A* 7, 16867-16875 (2019).
13. Zhang, S. et al. Amorphous and porous tungsten oxide films for fast-switching dual-band electrochromic smart windows. *Adv. Optical Mater.* 11, 2202115 (2023).
14. Li, Z. et al. Efficient electrochromic efficiency and stability of amorphous/crystalline tungsten oxide film. *J. Alloys Compd* 930, 167405 (2023).
15. Wang, J. et al. Amorphous mixed-vanadium-tungsten oxide films as optically passive ion storage materials for solid-state near-infrared electrochromic devices. *ACS Appl. Mater. Interfaces* 15, 7120-7128 (2023).

Doping :

16. Zhan, Y. et al. Ti-doped WO₃ synthesized by a facile wet bath method for improved electrochromism. *J. Mater. Chem. C* 5, 9995-10000 (2017).
17. Koo, B.-R., Kim, K.-H. & Ahn, H.-J. Switching electrochromic performance improvement enabled by highly developed mesopores and oxygen vacancy defects of Fe-doped WO₃ films. *Appl. Surf. Sci.* 453, 238-244 (2018).
18. Wang, W. et al. Niobium doped tungsten oxide mesoporous film with enhanced electrochromic and electrochemical energy storage properties. *J. Colloid. Interf. Sci* 535, 300-307 (2019).
19. Zhou, J., Wei, Y., Luo, G., Zheng, J. & Xu, C. Electrochromic properties of vertically aligned Ni-doped WO₃ nanostructure films and their application in complementary electrochromic devices. *J. Mater. Chem. C* 4, 1613-1622 (2016).

Surface-decorated :

20. Xu, J. et al. Electrochromic-tuned plasmonics for photothermal sterile window. *ACS Nano* 12, 6895-6903 (2018).

Figure R4 (Supplementary Figure 34). **a**, Summary of thermochromic performance (visible transmittance and solar modulation) in some of the best reported and selected works. **b**, The radar plot in which the visible transmittance, solar modulation, energy-saving efficiency, cyclic stability, and scalability of VO₂-based thermochromic windows are compared (Doping: Ref 24, Multilayer: Ref 30, Composite: Ref 38, Porous & Grid: Ref 41, Multiple stimulus: Ref 46).

Reference

Doping:

21. Wang, N., Liu, S., Zeng, X., Magdassi, S. & Long, Y. Mg/W-codoped vanadium dioxide thin films with enhanced visible transmittance and low phase transition temperature. *J. Mater. Chem. C* 3, 6771-6777 (2015).
22. Shen, N. et al. The synthesis and performance of Zr-doped and W-Zr-codoped VO₂ nanoparticles and derived flexible foils. *J. Mater. Chem. A* 2, 15087-15093 (2014).
23. Zhang, Z. et al. Thermochromic VO₂ thin films: Solution-based processing, improved optical properties, and lowered phase transformation temperature. *Langmuir* 26, 10738-10744 (2010).
24. Dai, L. et al. F-doped VO₂ nanoparticles for thermochromic energy-saving foils with modified color and enhanced solar-heat shielding ability. *Phys. Chem. Chem. Phys.* 15, 11723-11729 (2013).
25. Chen, S. et al. The visible transmittance and solar modulation ability of VO₂ flexible foils simultaneously improved by Ti doping: An optimization and first principle study. *Phys. Chem. Chem. Phys.* 15, 17537-17543 (2013).
26. Dietrich, M. K. et al. Influence of doping with alkaline earth metals on the optical properties of thermochromic VO₂. *J. Appl. Phys.* 117, 185301 (2015).

Multilayer:

27. Zhang, Z. et al. Solution-based fabrication of vanadium dioxide on f: SnO₂ substrates with largely enhanced thermochromism and low-emissivity for energy-saving applications. *Energy Environ. Sci.* 4, 4290-4297 (2011).
28. Liu, C. et al. Index-tunable anti-reflection coatings: Maximizing solar modulation ability for vanadium dioxide-based smart thermochromic glazing. *J. Alloys Compd* 731, 1197-1207 (2018).
29. Chen, Z. et al. VO₂-based double-layered films for smart windows: Optical design, all-solution preparation and improved properties. *Energy Mater. Sol. Cells* 95, 2677-2684 (2011).
30. Chang, T. et al. Mitigating deterioration of vanadium dioxide thermochromic films by interfacial encapsulation. *Matter* 1, 734-744 (2019).
31. Liu, C., Wang, N. & Long, Y. Multifunctional overcoats on vanadium dioxide thermochromic thin films with enhanced luminous transmission and solar modulation, hydrophobicity and anti-oxidation. *Appl. Surf. Sci.* 283, 222-226 (2013).
32. Hao, Q. et al. VO₂/TiN plasmonic thermochromic smart coatings for room-temperature applications. *Adv. Mater.* 30, 1705421 (2018).

Composite:

33. Gao, Y. et al. Enhanced chemical stability of VO₂ nanoparticles by the formation of SiO₂/VO₂ core/shell structures and the application to transparent and flexible VO₂-based composite foils with excellent thermochromic properties for solar heat control. *Energy Environ. Sci.* 5, 6104-6110 (2012).
34. Zhu, J. et al. Vanadium dioxide nanoparticle-based thermochromic smart coating: High luminous transmittance, excellent solar regulation efficiency, and near room temperature phase transition. *ACS Appl. Mater. Interfaces* 7, 27796-27803 (2015).
35. Liu, C. et al. VO₂/Si-Al gel nanocomposite thermochromic smart foils: Largely enhanced luminous transmittance and solar modulation. *J. Colloid Interface Sci.* 427, 49-53 (2014).
36. Chen, Z., Cao, C., Chen, S., Luo, H. & Gao, Y. Crystallised mesoporous TiO₂(a)-VO₂ (M/R) nanocomposite films with self-cleaning and excellent thermochromic properties. *J. Mater. Chem. A* 2, 11874-11884 (2014).
37. Zhu, J. et al. Hybrid films of VO₂ nanoparticles and a nickel (ii)-based ligand exchange thermochromic system: Excellent optical performance with a temperature responsive colour change. *New J. Chem.* 41, 830-835 (2017).
38. Moot, T., Palin, C., Mitran, S., Cahoon, J. F. & Lopez, R. Designing plasmon-enhanced thermochromic films using a vanadium dioxide nanoparticle elastomeric composite. *Adv. Optical Mater.* 4, 578-583 (2016).

Porous & Grid:

39. Ke, Y. et al. Two-dimensional SiO₂/VO₂ photonic crystals with statically visible and dynamically infrared modulated for smart window deployment. *ACS Appl. Mater. Interfaces* 8, 33112-33120 (2016).
40. Cao, X. et al. Nanoporous thermochromic VO₂(M) thin films: Controlled porosity, largely enhanced luminous transmittance and solar modulating ability. *Langmuir* 30, 1710-1715 (2014).
41. Zhuang, B. et al. 3D ordered macroporous VO₂ thin films with an efficient thermochromic modulation capability for advanced smart windows. *Adv. Optical Mater.* 7, 1900600 (2019).
42. Zhou, C. et al. 3D printed smart windows for adaptive solar modulations. *Adv. Optical Mater.* 8, 2000013 (2020).
43. Ke, Y. et al. Cephalopod-inspired versatile design based on plasmonic VO₂ nanoparticle for energy-efficient mechano-thermochromic windows. *Nano Energy* 73, 104785 (2020).

Electro-Thermochromic:

44. Shen, N. et al. Joule heating driven infrared switching in flexible VO₂ nanoparticle films with reduced energy consumption for smart windows. *J. Mater. Chem. A* 7, 4516-4524 (2019).
45. Chen, S. et al. Gate-controlled VO₂ phase transition for high-performance smart windows. *Sci. Adv.* 5, eaav6815 (2019).
46. Ke, Y. et al. Adaptive thermochromic windows from active plasmonic elastomers. *Joule* 3, 858-871 (2019).

In tables 8& 9 (Section 17) of the supplementary Information, the authors presented the cost of production of the SLE4, and TCM4 films. While it is a bit confusing what they referred to by “fabrication costs of the materials in the unit area of SLE4 film”, I suppose the labor cost for fabricating the films is not considered.

Thank you for the kind suggestion. The labor cost for fabricating the films has been added in **Table R1 and R2 (Supplementary Table 9 and 10). According to the hourly wage standards in Hefei published by the Ministry of Human Resources and Social Security of the People's Republic of China (http://www.mohrss.gov.cn/SYrlzyhshbzb/laodongguanxi_/fwyd/202301/t20230102_492654.html), one person would spend 2.5 h to fabricate SLE4 film and the labor cost was 5.96 dollars h⁻¹ in Hefei. As for the TCM4 film, one person would spend 1 h to complete it and the labor cost was 2.384 dollars h⁻¹ in Hefei.

Table R1 (Supplementary Table 11). The fabrication costs of the materials in the unit area of SLE4 film.

	Au NRs	Ag NWs	W₁₈O₄₉ NWs	DMF+ CHCl₃	Substrate	labor cost	Total
Unit price	5.23 dollar/m ²	0.079 dollar/m ²	22.08 dollar/m ²	0.25 dollar/m ²	1.583 dollar/m ²	5.96 dollar/m ²	35.182 dollar/m ²

Table R2 (Supplementary Table 12). The fabrication costs of the materials in the unit area of TCM4 film.

	W-VO₂-1 NWs	W-VO₂-2 NWs	W-VO₂-3 NWs	DMF+ CHCl₃	Substrate	labor cost	Total
Unit price	0.098 dollar/m ²	0.098 dollar/m ²	0.098 dollar/m ²	0.25 dollar/m ²	1.589 dollar/m ²	2.384 dollar/m ²	4.511 dollar/m ²

Reviewer #2 (Remarks to the Author):

The paper entitled “Nanowire-Based Smart Windows for Solar Radiation Regulation” shows solar radiation regulation by using plasmonic Au nanorods, electrochromic tungsten oxide nanowires and thermochromic Vanadium oxide nanowires. The authors propose new ideas of co-assembling different optical materials to have a high performance of smart window.

This manuscript shows detail characterizations of optical materials for smart window applications. There are some suggestions to improve the quality of the paper.

1. The authors suggest importance of modulation of sunlight at special band ranges. It is very good to have a detail description or demonstration of the modulation for some specific applications. The reader may not understand the point of this paper without proper demonstrations.

****Thank you for the kind suggestion. The visible light (VIS) and near-infrared (NIR) spectra have significantly different functions and need to be separated regulation in buildings, as the VIS contains visibility, color, and heat, and the NIR is only for heat (*ACS Materials Lett.* 2020, 2, 1624; *Annu. Rev. Chem. Biomol. Eng.* 2016, 7, 283). The ability to modulate the NIR transmission through the windows has a significant effect on indoor temperature and energy saving. However, conventional smart windows usually block all wavelengths of sunlight for a good cooling effect. As a result, extra energy is needed for constant indoor lighting and activating windows. Therefore, the new smart windows need to block near-infrared light while not interfering too much with visible light transmission. Especially the near-infrared wavelengths regions of 760-1360 nm (NIR-1), which accounts for 36% of the solar radiation, and the near-infrared wavelengths region between 1360-2500 nm (NIR-2) just accounts for 11% of the solar radiation.**

In general, glass with an average visible transmission value above 60% looks clear, and any value below 50% begins to look dark, colored, and/or reflective (*Nat. Energy* 2017, 2, 849). In **Figure R5 (Supplementary Fig. 13)** and **Table R3 (Supplementary Table 7)**, the SLE4 film with Au NRs of different sizes could selectively block more than 50% of the NIR-1 light (760-1360 nm) without energy input, and 70% transmittance is retained in the visible region to meet daylighting requirements. While the EC film without Au NRs requires a certain voltage to achieve a similar level of modulation as the

SLE4 film. In the colored state (EC colored-1), the EC film makes more near-infrared light passes through to ensure a sufficient amount of visible light. In the other colored state (EC colored-2), the EC film absorbs a portion of visible light to achieve the level of SLE4 film for near-infrared light absorption. Thus, the selective absorption function of the film could continuously block near-infrared light without activating the electrochromic part, which meets the daylighting requirements and reduces the dependence of modulation performance on electrochromic function.

Enough visible light provides indoor lighting and temperature, making the living environment more comfortable. It is essential to save energy and create a comfortable living environment by modulating sunlight at special band ranges.

Figure R5 (Supplementary Figure 13). **a**, The transmittance spectra of SLE4 and EC at different colored states. **b**, The transmitted solar irradiance of SLE4 and EC at different colored states.

Table R3 (Supplementary Table 7). Integrated optical transmittance (T) and solar irradiance transmittance (T') of EC at bleached and colored states, compared to the SLE4 films in the VIS (400-760 nm), NIR-1 (760-1360 nm) regions.

	T_{VIS}	$T_{\text{NIR-1}}$	T'_{VIS}	$T'_{\text{NIR-1}}$
EC bleached	88.3%	81.8%	87.8%	82.2%
EC colored-1	71.0%	59.3%	70.9%	59.7%
EC colored-2	64.5%	47.9%	64.2%	48.4%
SLE4	70.3%	48.1%	70.5%	49.0%

2. Please show reliability tests of the devices at real ambient conditions.

Thank you for the kind comments. We placed the model house with SLE4 and WRT4 smart windows at real ambient conditions for a long time, and the temperature of the blackbody was recorded to verify the stability of the cooling performance of SLE4 and WRT4 films. **In Figure R6 (Supplementary Figure 17) and R7 (Supplementary Figure 30), Due to the stability of the Au NRs and the overall device, the cooling performance of the SLE4 film shows almost no degradation even after 60 days. While the cooling performance of the WRT4 film has diminished by 0.6 °C at the same time, this is because the thermochromic properties of the W-VO₂ NWs are weakened by partial oxidation in the air.

Figure R6 (Supplementary Figure 17). The stability of the cooling performance of SLE4 film.

Figure R7 (Supplementary Figure 30). The stability of the cooling performance of WRT4 film.

Reviewer #3 (Remarks to the Author):

The authors presented a generalized strategy to fabricate nanowire-based smart windows to modulate the solar heat gain with multispectral controllability. This work shows beautifully made nanowires with thorough characterization. The ordered structure produced by the Langmuir-Blodgett method is interesting and effective for improving the visual appearance (low haze). Still, I would request a few additional information and clarification before recommending for publication:

1. All of the nanowires used in this work were synthesized by following previous works, as noted in the manuscript's reference list. Moreover, their applications were the same – WO_x for electrochromics, W-VO₂ for thermochromics, AgNWs for transparent conductors, and AuNRs for plasmonic absorption.

After reading through the manuscript, I was still trying to grasp the novelty of this work, except for the L-B fabrication. For the readers of Nature Communications, it is essential to explain the novelty by providing the comparison chart with relevant prior works.

**** Thank you for the kind comments. We apologize for not accurately conveying our innovation to reviewers through our inaccurate descriptions and immature writing. In the new version of the manuscript, we pay more attention to the novelty description. Co-assembling multiple optical materials into ordered structures is a new and effective way to optimize the performance of smart windows, and the improved modulation performance stems from the synergistic effect of the multiple optical materials. Most scientists fabricate smart windows by developing new chromogenic materials or designing complex optical structures. Contrary to that, we prepare smart windows by co-assembling multiple types of optical materials, and the performance of the devices is easily tuned by adjusting the specific composition and structure of these materials.**

Here, we summarize the novelty and significance of this work as follows:

(1) To date, multifarious methods are mostly focused on the design of smart windows with novel chromogenic materials or complex optical structures, whereas little attention is paid to regulating the components and structures of the smart windows. Herein, we fabricate the electrochromic and thermochromic smart windows with tunable components and ordered structures, and the Langmuir-Blodgett (LB) co-assembly technology is simple and efficient for modulating the components and structure of devices.

To tackle the bottleneck of conventional electrochromic windows, Au NRs were introduced into the electrochromic window to continuously absorb near-infrared light without affecting visible light transmission too much, and the absorption range can be broadened by co-assembling Au NR mixture with different aspect ratios. The electrochromic function of $W_{18}O_{49}$ NWs is used as an active option for further modulation of solar radiation according to the wishes of the households and the actual weather.

To improve the modulation performance of typical thermochromic windows based on VO_2 , the thermochromic $W-VO_2$ NWs with different doping amounts are co-assembled and extended fixed response temperature value to a wide temperature range of 30 °C~50 °C. The wide response range allows smart windows to progressively adjust their blocking ability as the temperature changes, dynamically regulating the room temperature in various real-world weather conditions.

(2) The haze of smart windows can be significantly reduced by sequencing the orientation of the optical materials during the co-assembly process without complex structural design.

(3) The simplicity of the co-assembly method enables the easy fabrication of large-area smart windows with dimensions of 25×20 cm².

(4) The energy saving energy-saving simulation shows the promising energy-saving performance of smart windows in different cities.

It is worth mentioning that the strategy here shows a superposition effect, which means smart windows design can be pushed forward further combined with novel chromogenic materials. In **Figure R8 and R9 (Supplementary Figure 33 and 34)** Compared with the previous work, the electrochromic and thermochromic smart windows prepared by our LB co-assembly strategy show significant improvements in sunlight modulation and cooling effect, this simplicity and versatility of the co-assembly method allows for the large-scale preparation of smart windows.

Figure R8 (Supplementary Figure 33). **a**, Summary of electrochromic performance (transmittance modulation and switching time) in some of the best reported and selected works. **b**, The radar plot in which the transmittance modulation, switching time, coloration efficiency, cyclic stability, and scalability of Tungsten Oxide based thermochromic windows are compared (Nanostructure: Ref 1, Multilayer: Ref 9, Amorphous: Ref 11, Doping: Ref 16, Surface-decorated: Ref 20).

Reference

Nanostructure:

1. Gu, H. *et al.* Highly efficient, near-infrared and visible light modulated electrochromic devices based on polyoxometalates and $W_{18}O_{49}$ nanowires. *ACS Nano* **12**, 559-567 (2018).
2. Wang, J.-L., Lu, Y.-R., Li, H.-H., Liu, J.-W. & Yu, S.-H. Large area co-assembly of nanowires for flexible transparent smart windows. *J. Am. Chem. Soc.* **139**, 9921-9926 (2017).
3. Heo, S., Kim, J., Ong, G. K. & Milliron, D. J. Template-free mesoporous electrochromic films on flexible substrates from tungsten oxide nanorods. *Nano Lett.* **17**, 5756-5761 (2017).
4. Li, H., McRae, L. & Elezzabi, A. Y. Solution-processed interfacial PEDOT:PSS assembly into porous tungsten molybdenum oxide nanocomposite films for electrochromic applications. *ACS Appl. Mater. Interfaces* **10**, 10520-10527 (2018).
5. Nguyen, T. D. *et al.* Efficient near infrared modulation with high visible transparency using SnO_2 - WO_3 nanostructure for advanced smart windows. *Adv. Optical Mater.* **7**, 1801389 (2019).
6. Qu, H. *et al.* Highly robust and flexible $WO_3 \cdot 2H_2O$ / PEDOT films for improved electrochromic performance in near-infrared region. *Sol. Energy Mater Sol. Cells* **163**, 23-30 (2017).

Multilayer :

7. Li, H., Lv, Y., Zhang, X., Wang, X. & Liu, X. High-performance ITO-free electrochromic films based on bi-functional stacked $WO_3/Ag/WO_3$ structures. *Sol. Energy Mater Sol. Cells* **136**, 86-91 (2015).
8. Xiao, L., Lv, Y., Dong, W., Zhang, N. & Liu, X. Dual-functional WO_3 nanocolumns with broadband antireflective and high-performance flexible electrochromic properties. *ACS Appl. Mater. Interfaces* **8**, 27107-27114 (2016).
9. Najafi-Ashtiani, H., Akhavan, B., Jing, F. & Bilek, M. M. Transparent conductive dielectric-metal-dielectric structures for electrochromic applications fabricated by high-power impulse magnetron sputtering. *ACS Appl. Mater. Interfaces* **11**, 14871-14881 (2019).
10. Dong, W. *et al.* Bifunctional MoO_3 - $WO_3/Ag/MoO_3$ - WO_3 films for efficient ITO-free electrochromic devices. *ACS Appl. Mater. Interfaces* **8**, 33842-33847 (2016).

Amorphous:

11. Cheng, W. *et al.* Photodeposited amorphous oxide films for electrochromic windows. *Chem* **4**, 821-832 (2018).
12. Huo, X. *et al.* Bifunctional aligned hexagonal/amorphous tungsten oxide core/shell nanorod arrays with enhanced electrochromic and pseudocapacitive performance. *J. Mater. Chem. A* **7**, 16867-16875 (2019).

13. Zhang, S. et al. Amorphous and porous tungsten oxide films for fast-switching dual-band electrochromic smart windows. *Adv. Optical Mater.* 11, 2202115 (2023).
14. Li, Z. et al. Efficient electrochromic efficiency and stability of amorphous/crystalline tungsten oxide film. *J. Alloys Compd* 930, 167405 (2023).
15. Wang, J. et al. Amorphous mixed-vanadium-tungsten oxide films as optically passive ion storage materials for solid-state near-infrared electrochromic devices. *ACS Appl. Mater. Interfaces* 15, 7120-7128 (2023).

Doping :

16. Zhan, Y. et al. Ti-doped WO₃ synthesized by a facile wet bath method for improved electrochromism. *J. Mater. Chem. C* 5, 9995-10000 (2017).
17. Koo, B.-R., Kim, K.-H. & Ahn, H.-J. Switching electrochromic performance improvement enabled by highly developed mesopores and oxygen vacancy defects of Fe-doped WO₃ films. *Appl. Surf. Sci.* 453, 238-244 (2018).
18. Wang, W. et al. Niobium doped tungsten oxide mesoporous film with enhanced electrochromic and electrochemical energy storage properties. *J. Colloid. Interf. Sci* 535, 300-307 (2019).
19. Zhou, J., Wei, Y., Luo, G., Zheng, J. & Xu, C. Electrochromic properties of vertically aligned Ni-doped WO₃ nanostructure films and their application in complementary electrochromic devices. *J. Mater. Chem. C* 4, 1613-1622 (2016).

Surface-decorated :

20. Xu, J. et al. Electrochromic-tuned plasmonics for photothermal sterile window. *ACS Nano* 12, 6895-6903 (2018).

Figure R9 (Supplementary Figure 34). **a**, Summary of thermochromic performance (visible transmittance and solar modulation) in some of the best reported and selected works. **b**, The radar plot in which the visible transmittance, solar modulation, energy-saving efficiency, cyclic stability, and scalability of VO₂-based thermochromic windows are compared (Doping: Ref 24, Multilayer: Ref 30, Composite: Ref 38, Porous & Grid: Ref 41, Multiple stimulus: Ref 46).

Reference

Doping:

21. Wang, N., Liu, S., Zeng, X., Magdassi, S. & Long, Y. Mg/W-codoped vanadium dioxide thin films with enhanced visible transmittance and low phase transition temperature. *J. Mater. Chem. C* 3, 6771-6777 (2015).
22. Shen, N. et al. The synthesis and performance of Zr-doped and W-Zr-codoped VO₂ nanoparticles and derived flexible foils. *J. Mater. Chem. A* 2, 15087-15093 (2014).
23. Zhang, Z. et al. Thermochromic VO₂ thin films: Solution-based processing, improved optical properties, and lowered phase transformation temperature. *Langmuir* 26, 10738-10744 (2010).

24. Dai, L. et al. F-doped VO₂ nanoparticles for thermochromic energy-saving foils with modified color and enhanced solar-heat shielding ability. *Phys. Chem. Chem. Phys.* 15, 11723-11729 (2013).
25. Chen, S. et al. The visible transmittance and solar modulation ability of VO₂ flexible foils simultaneously improved by Ti doping: An optimization and first principle study. *Phys. Chem. Chem. Phys.* 15, 17537-17543 (2013).
26. Dietrich, M. K. et al. Influence of doping with alkaline earth metals on the optical properties of thermochromic VO₂. *J. Appl. Phys.* 117, 185301 (2015).

Multilayer:

27. Zhang, Z. et al. Solution-based fabrication of vanadium dioxide on f: SnO₂ substrates with largely enhanced thermochromism and low-emissivity for energy-saving applications. *Energy Environ. Sci.* 4, 4290-4297 (2011).
28. Liu, C. et al. Index-tunable anti-reflection coatings: Maximizing solar modulation ability for vanadium dioxide-based smart thermochromic glazing. *J. Alloys Compd* 731, 1197-1207 (2018).
29. Chen, Z. et al. VO₂-based double-layered films for smart windows: Optical design, all-solution preparation and improved properties. *Energy Mater. Sol. Cells* 95, 2677-2684 (2011).
30. Chang, T. et al. Mitigating deterioration of vanadium dioxide thermochromic films by interfacial encapsulation. *Matter* 1, 734-744 (2019).
31. Liu, C., Wang, N. & Long, Y. Multifunctional overcoats on vanadium dioxide thermochromic thin films with enhanced luminous transmission and solar modulation, hydrophobicity and anti-oxidation. *Appl. Surf. Sci.* 283, 222-226 (2013).
32. Hao, Q. et al. VO₂/TiN plasmonic thermochromic smart coatings for room-temperature applications. *Adv. Mater.* 30, 1705421 (2018).

Composite:

33. Gao, Y. et al. Enhanced chemical stability of VO₂ nanoparticles by the formation of SiO₂/VO₂ core/shell structures and the application to transparent and flexible VO₂-based composite foils with excellent thermochromic properties for solar heat control. *Energy Environ. Sci.* 5, 6104-6110 (2012).
34. Zhu, J. et al. Vanadium dioxide nanoparticle-based thermochromic smart coating: High luminous transmittance, excellent solar regulation efficiency, and near room temperature phase transition. *ACS Appl. Mater. Interfaces* 7, 27796-27803 (2015).
35. Liu, C. et al. VO₂/Si-Al gel nanocomposite thermochromic smart foils: Largely enhanced luminous transmittance and solar modulation. *J. Colloid Interface Sci.* 427, 49-53 (2014).
36. Chen, Z., Cao, C., Chen, S., Luo, H. & Gao, Y. Crystallised mesoporous TiO₂(a)-VO₂ (M/R) nanocomposite films with self-cleaning and excellent thermochromic properties. *J. Mater. Chem. A* 2, 11874-11884 (2014).
37. Zhu, J. et al. Hybrid films of VO₂ nanoparticles and a nickel (ii)-based ligand exchange thermochromic system: Excellent optical performance with a temperature responsive colour change. *New J. Chem.* 41, 830-835 (2017).
38. Moot, T., Palin, C., Mitran, S., Cahoon, J. F. & Lopez, R. Designing plasmon-enhanced thermochromic films using a vanadium dioxide nanoparticle elastomeric composite. *Adv. Optical Mater.* 4, 578-583 (2016).

Porous & Grid:

39. Ke, Y. et al. Two-dimensional SiO₂/VO₂ photonic crystals with statically visible and dynamically infrared modulated for smart window deployment. *ACS Appl. Mater. Interfaces* 8, 33112-33120 (2016).
40. Cao, X. et al. Nanoporous thermochromic VO₂(M) thin films: Controlled porosity, largely enhanced luminous transmittance and solar modulating ability. *Langmuir* 30, 1710-1715 (2014).
41. Zhuang, B. et al. 3D ordered macroporous VO₂ thin films with an efficient thermochromic modulation capability for advanced smart windows. *Adv. Optical Mater.* 7, 1900600 (2019).
42. Zhou, C. et al. 3D printed smart windows for adaptive solar modulations. *Adv. Optical Mater.* 8, 2000013 (2020).
43. Ke, Y. et al. Cephalopod-inspired versatile design based on plasmonic VO₂ nanoparticle for energy-efficient mechano-thermochromic windows. *Nano Energy* 73, 104785 (2020).

Electro-Thermochromic:

44. Shen, N. et al. Joule heating driven infrared switching in flexible VO₂ nanoparticle films with reduced energy consumption for smart windows. *J. Mater. Chem. A* 7, 4516-4524 (2019).
45. Chen, S. et al. Gate-controlled VO₂ phase transition for high-performance smart windows. *Sci. Adv.* 5, eaav6815 (2019).
46. Ke, Y. et al. Adaptive thermochromic windows from active plasmonic elastomers. *Joule* 3, 858-871 (2019).

2. One potential advantage of W-VO₂ nanowires is to provide “wide-range”. While this seems to be a new function, I wonder what practical scenario would favor a gradual change versus a sharp one? The nanowires block only NIR light, so there isn’t a concern about changing visual appearance. Shouldn’t a sharp and large transition save more HVAC energy than a gradual one?

**Thank you for the critical question. Buildings have different needs for sunlight in distinct seasons, especially in the cold winter and hot summer, while the single type of W-VO₂ with a sharp phase transition is proved to be unsuitable for seasonal changes. On the one hand, the thermochromic windows with lower response temperature are suitable for hot summer, but block some of the sunlight in other seasons and cause unnecessary energy consumption. On the other hand, the thermochromic windows with higher response temperature can transmit sunlight as much as possible in cold winter, while the cooling effect is not obvious in summer. The thermochromic windows with a wide temperature stimulus-response range could realize the desire for progressive modulation of sunlight, that is different doping amounts of W-VO₂ nanowires undergoing phase transition are different over a broad temperature interval.

In **Figure R10 (Fig. 4c)** and **R11 (Supplementary Figure 21)**, we have calculated the specific transmittance of TC and WRT films based on their spectral changes at different temperatures (**Table R5 (Supplementary Table 9)**). Smart windows with a wide temperature stimulus-response range (WRT) are more suitable for the actual temperature change than thermochromic windows (TC) containing only one type of nanowires. Specifically, the WRT4 window does not block sunlight too much when the temperature is below 30 °C. A certain amount of nanowires undergo a phase transition within 30 °C~40 °C, and the blocking ability is further enhanced as the temperature rises to 50 °C. This is also reflected in the results of the simulated light test, **Figure R12 (Fig. 4f)** shows the temperature change of the blackbody in the model chamber installed with glass, TC, and WRT films under the simulated illumination. At low light intensity (25 and 50 mW cm⁻²), the cooling performance of WRT4 is close to that of TC1 and TC2 with high phase change temperature, which means that more sunlight will enter the room through the smart windows on cold days. At high light intensity (75 and 100 mW cm⁻²), the cooling effect of WRT4 is close to that of TC3 with low phase change temperature, which indicates that more sunlight will be blocked in hot weather.

Figure R10 (Fig. 4c). The transmittance spectra and the transmitted solar irradiance of WRT4 film at different temperatures.

Figure R11 (Supplementary Figure 21). a, b, c, The transmittance spectra of TC1, TC2, and TC3 films. d, e, f, The transmittance spectra of WRT1, WRT2, and WRT3 films at different temperature.

Table R5 (Supplementary Table 9). Integrated optical transmittance (T) of the TC and WRT films in the VIS (400-760 nm), NIR-1 (760-2500 nm) regions at different temperature.

	VIS	TC1	TC2	TC3	WRT1	WRT2	WRT3	WRT4
NIR-1								
20 °C		60.0%/	66.1%/	61.2%/	65.7%/	62.0%/	62.9%/	65.9%/
		85.7%	82.2%	84.3%	84.5%	83.5%	81.7%	87.5%
30 °C		60.1%/	66.2%/	61.8%/	66.1%/	62.0%/	62.9%/	66.2%/
		85.4%	80.6%	80.3%	84.2%	82.1%	79.0%	86.4%
40 °C		60.2%/	66.3%/	61.8%/	65.3%/	62.1%/	62.9%/	66.3%/
		84.8%	79.3%	71.5%	73.8%	75.7%	75.3%	78.0%
50 °C		60.2%/	66.5%/	63.7%/	65.1%/	64.2%/	63.0%/	66.6%/
		65.8%	67.6%	64.4%	67.1%	65.6%	68.9%	62.6%

Figure R12 (Fig. 4f). The temperature change of blackbody in the model chamber installed with glass, TC and WRT films under the simulated illumination of different intensities.

3. Based on the information provided in the current manuscript, it is difficult to interpret the temperature rise in the simulated house experiments. The optical measurement (transmittance) makes perfect sense because it is well-defined. However, the temperature rise is dependent not only on the window but the chamber and ambient convective heat transfer. How do we link a “2 degree Celsius drop” to cooling power or energy saving when it comes to HVAC thermal engineering?

Thank you for the question. The simulated light tests are done in a dark, closed room (room temperature is kept at 26 °C) to avoid the influence of external factors on the experimental results. To link a “2 degree Celsius drop” to cooling power or energy saving when it comes to HVAC thermal engineering, and further investigate the energy-saving performance of our smart windows. An energy-saving simulation was performed, which designs the actual-size building model in the *EnergyPlus* simulation software. In **Figure R13 (Supplementary Figure 32), an $8 \times 6 \times 2.7$ m³ model house was built with two 5×2 m² windows and two 3×2 m² windows on four walls. Climate data of Riyadh (Saudi Arabia) and Hong Kong (China) were selected to analyze the window performance. In the simulation, the electrochromic function of SLE window is activated when the temperature is above 30 °C, and the modulation of WRT window is dynamically adjusted with ambient temperature. According to the specific transmittance of smart windows at different temperature, the solar radiation entering the room is dynamically changed in the simulation, and we can get the final specific energy saving data.

Figure R14 (Fig. 5d-f) describes the monthly total energy load of the bare glass, SLE, and WRT smart windows in Riyadh (Saudi Arabia) and Hong Kong (China). The monthly total energy load (including the HVAC system and lighting system) of the building is significantly reduced by using these smart windows, especially in the hot summer months. Moreover, we calculated the energy saving by plotting the energy consumption difference between the smart windows and normal glass windows. In particular, the SLE window can reduce 16.3% and 19.6% total energy consumption in Riyadh and Hong Kong, respectively. As for WRT window, 5.2% and 6.7% of total energy consumption decreased in Riyadh and Hong Kong, respectively. The simulation results indicate that the SLE and WRT smart windows show promising energy-saving performance in different cities.

Figure R13 (Supplementary Figure 32). Schematic diagram of the building model used in the simulation.

Figure R14 (Fig. 5d-f). **a**, Monthly total energy load of the bare glass, SLE and WRT smart windows in Riyadh (Saudi Arabia). **b**, Monthly total energy load of the normal glass, SLE and WRT smart windows in Hong Kong (China). **c**, Calculated total energy load per year for the windows applied in Riyadh and Hong Kong, respectively. The insert is the total energy saving by using SLE and WRT smart window over a normal glass window.

Reviewer #4 (Remarks to the Author):

This manuscript has been focused on the fabrication of wavelength conversion material for solar light. Smart function of it has not been clarified based on the mechanism and basic principle on the oxidation state change of the ionic elements and the quantitative analysis of the transmittance and conversion efficiency were not supplied. Authors should define the basic principles of the wavelength conversion by the changing parameters of the W ion and quantitative analysis of the solar light wavelength change by determining transmittance at specific wavelength range. This is critical data for the wavelength conversion efficiency of the solar light regulation. So this manuscript should be rejected as the present form. Authors must add more required specific data to publish this manuscript.

****Thank you for the kind comments. We apologize for the misunderstanding caused to reviewers by our inaccurate descriptions and immature writing. The phase transition temperature of VO₂ nanowires is regulated by doping with different amounts of W elements, and the modulation range of W- VO₂ NWs with different phase transition temperature is mainly in the near-infrared region and does not have a specific spectral selectivity in the 760-1360 nm region (*Nat. Commun.* 2020, 11, 3591; *Adv. Optical Mater.* 2022, 10, 2201326; *J. Am. Chem. Soc.* 2013, 135, 4850; *J. Mater. Chem.* 2011, 21, 5580).**

In the revised version, we have rewritten the description of the preparation process and regulation mechanism of the devices. The novel electrochromic and thermochromic smart windows with tunable components and ordered structures are prepared by a co-assembly strategy, respectively.

In **Figure R15**, Different aspect ratios of Au NRs with narrow absorption range are synthesized and mixed to cover the 760-1360 nm band, then the multi-sized Au NR mixture, electrochromic $W_{18}O_{49}$ NWs, and conductive Ag NWs are co-assembled into an ordered network-like structure as selective light absorption electrochromic (SLE) smart window. This window could selectively absorb near-infrared light and avoid the high dependence on electrochromic functions. When the negative voltage is applied, the device transitions to a colored state that can further block the majority of near-infrared light.

In **Figure R16**, The thermochromic windows based on one type of $W-VO_2$ NWs have a narrow temperature response range and are not suitable for actual weather changes. Compared with the fixed phase change temperature (68 °C) of VO_2 , the thermochromic $W-VO_2$ NWs with different tungsten (W) element doping are co-assembled to extend fixed response temperature value to a wide temperature range of 30 °C~50 °C. At low temperature, only the nanowires with high doping amounts undergo a phase change, while several nanowires with different doping amounts are in the phase transition state with the increase in temperature. This wide response range allows smart windows to progressively adjust their blocking ability as the temperature changes.

Figure R15. The schematic illustration for the fabrication and modulation mechanism of electrochromic film with selective light absorption.

Figure R16. The schematic illustration for the fabrication and modulation mechanism of thermochromic film with a wide response-range.

And we have modified the schematic illustration for the fabrication and modulation mechanism of smart windows in **Figure R17 (Fig. 1)** to allow reviewers to fully understand the mechanism of modulation.

Figure R17 (Fig. 1). The schematic illustration for the fabrication and modulation mechanism of smart windows. **a**, The preparation strategy of selective light absorption electrochromic (SLE) smart window based on co-assembly of multiple nanowires and Au nanorods. **b**, The preparation strategy of wide-range thermochromic (WRT) smart window based on co-assembly of nanowires with different doping amounts. **c**, The working effect of the house equipped with these smart windows when applied with a small voltage or ambient temperature change.

To quantitatively analyze the transmittance variation and the modulation of solar radiation in the WRT films, we have calculated the optical transmittance and solar irradiance transmittance of TC and WRT films based on their spectral changes at different temperatures. In **Table R6 and R7 (Supplementary Table 9 and 10)**, smart windows with a wide temperature stimulus-response range (WRT) are more suitable for the actual temperature change than thermochromic windows (TC) containing only one type of nanowires. Specifically, the WRT4 window does not block sunlight too much when the temperature is below 30 °C. A certain amount of nanowires undergo a phase transition within 30 °C~40 °C, and the blocking ability is further enhanced as the temperature rises to 50 °C. The higher the content of W-VO₂-3 NWs in the film is, the more suitable to block solar radiation at a lower temperature. The higher the content of W-VO₂-1 NWs in the film is, the more suitable to block solar radiation at a higher temperature.

Table R6 (Supplementary Table 9). Integrated optical transmittance of the TC and WRT films in the VIS (400-760 nm), NIR (760-2500 nm) regions at different temperature.

	VIS	TC1	TC2	TC3	WRT1	WRT2	WRT3	WRT4
NIR								
20 °C		60.0%/	66.1%/	61.2%/	65.7%/	62.0%/	62.9%/	65.9%/
		85.7%	82.2%	84.3%	84.5%	83.5%	81.7%	87.5%
30 °C		60.1%/	66.2%/	61.8%/	66.1%/	62.0%/	62.9%/	66.2%/

	85.4%	80.6%	80.3%	84.2%	82.1%	79.0%	86.4%
40 °C	60.2%/	66.3%/	61.8%/	65.3%/	62.1%/	62.9%/	66.3%/
	84.8%	79.3%	71.5%	73.8%	75.7%	75.3%	78.0%
50 °C	60.2%/	66.5%/	63.7%/	65.1%/	64.2%/	63.0%/	66.6%/
	65.8%	67.6%	64.4%	67.1%	65.6%	68.9%	62.6%

Table R7 (Supplementary Table 10). Solar irradiance transmittance of the TC and WRT films in the VIS (400-760 nm), NIR (760-2500 nm) regions at different temperature.

	VIS	TC1	TC2	TC3	WRT1	WRT2	WRT3	WRT4
NIR								
20 °C	59.7%/	65.9%/	60.9%/	65.7%/	61.7%/	62.8%/	65.6%/	
	78.5%	77.8%	78.5%	80.0%	77.9%	76.8%	82.2%	
30 °C	59.9%/	66.0%/	61.5%/	65.4%/	61.8%/	62.8%/	65.9%/	
	78.5%	76.9%	76.0%	79.7%	76.9%	74.8%	81.7%	
40 °C	59.9%/	66.0%/	61.5%/	64.7%/	61.8%/	62.7%/	66.0%/	
	78.2%	75.8%	71.7%	72.5%	72.3%	72.1%	76.0%	
50 °C	60.0%/	66.3%/	63.4%/	64.7%/	64.0%/	62.8%/	66.3%/	
	62.0%	69.0%	67.8%	68.1%	66.8%	68.1%	67.7%	

Reviewers' Comments:

Reviewer #1:

Remarks to the Author:

The authors have revised the manuscript considering all the comments of the reviewers. The revised manuscript is much better and easier to understand. The results are presented and discussed in a better way, allowing readers to understand the physical phenomena behind the associated effects.

Although there are still a few grammatical and typo mistakes remain in the manuscript, I consider the manuscript suitable for publication in Nature communications. Minor grammatical and typo mistakes can be taken care of during the production process.

Reviewer #2:

Remarks to the Author:

The authors reported several good points. It is worth to be published Nature Communications.

Reviewer #3:

Remarks to the Author:

I appreciate the authors' response to my questions. I will be happy to recommend it for publication.

Reviewer #4:

Remarks to the Author:

Title: Nanowire-Based Smart Windows for Solar Radiation Regulation
(Nature Communications)

Comments,

The revised version of the manuscript entitled "Nanowire-Based Smart Windows for Solar Radiation Regulation" is smartly modified with precise changes, and authors have given proper explanation for all the questions. I believe the work is now well organized and well written. Hence, I would like to recommend the paper for publication in Nature Communications.

Manuscript ID: NCOMMS-22-47814A-Z

“Nanowire-based smart windows combining electro- and thermochromic for dynamic regulation of solar radiation”

REVIEWERS' COMMENTS

Reviewer #1 (Remarks to the Author):

The authors have revised the manuscript considering all the comments of the reviewers. The revised manuscript is much better and easier to understand. The results are presented and discussed in a better way, allowing readers to understand the physical phenomena behind the associated effects.

****We thank the reviewer for strong support on the publication of this work.**

Although there are still a few grammatical and typo mistakes remain in the manuscript, I consider the manuscript suitable for publication in Nature communications. Minor grammatical and typo mistakes can be taken care of during the production process.

****Thank you for the kind comments. We have corrected these grammatical and typo mistakes remain in the manuscript.**

Reviewer #2 (Remarks to the Author):

The authors reported several good points. It is worth to be published Nature Communications.

****We thank the reviewer for strong support on the publication of this work.**

Reviewer #3 (Remarks to the Author):

I appreciate the authors' response to my questions. I will be happy to recommend it for publication.

****We thank the reviewer for strong support on the publication of this work.**

Reviewer #4 (Remarks to the Author):

Title: Nanowire-Based Smart Windows for Solar Radiation Regulation

(Nature Communications)

Comments,

The revised version of the manuscript entitled “Nanowire-Based Smart Windows for Solar Radiation Regulation” is smartly modified with precise changes, and authors have given proper explanation for all the questions. I believe the work is now well organized and well written. Hence, I would like to recommend the paper for publication in Nature Communications.

****We thank the reviewer for strong support on the publication of this work.**